# Revisiting, Benchmarking and Understanding Unsupervised Graph Domain Adaptation

**Meihan Liu[1], Zhen Zhang[2]\*, Jiachen Tang[1], Jiajun Bu[1], Bingsheng He[2], Sheng Zhou[1]**
[1]Zhejiang Key Laboratory of Accessible Perception and Intelligent Systems,
College of Computer Science, Zhejiang University [2]National University of Singapore
`{lmh_zju,tangjc,bjj,zhousheng_zju}@zju.edu.cn`
`zhen@nus.edu.sg, hebs@comp.nus.edu.sg`

## Abstract

Unsupervised Graph Domain Adaptation (UGDA) involves the transfer of knowledge from a label-rich source graph to an unlabeled target graph under domain discrepancies. Despite the proliferation of methods designed for this emerging task, the lack of standard experimental settings and fair performance comparisons makes it challenging to understand which and when models perform well across different scenarios. To fill this gap, we present the first comprehensive benchmark for unsupervised graph domain adaptation named GDABench, which encompasses 16 algorithms across diverse adaptation tasks. Through extensive experiments, we observe that the performance of current UGDA models varies significantly across different datasets and adaptation scenarios. Specifically, we recognize that when the source and target graphs face significant distribution shifts, it is imperative to formulate strategies to effectively address and mitigate graph structural shifts. We also find that with appropriate neighbourhood aggregation mechanisms, simple GNN variants can even surpass state-of-the-art UGDA baselines. To facilitate reproducibility, we have developed an easy-to-use library PyGDA for training and evaluating existing UGDA methods, providing a standardized platform in this community. Our source codes and datasets can be found at `https://github.com/pygda-team/pygda`.

## 1 Introduction

The last decade has witnessed significant advancements in Graph Neural Networks (GNNs) with their successful applications spanning various fields [1, 2], including social network analysis [3, 4], protein interaction prediction [5], and traffic flow forecasting [6], etc. However, the presence of distribution shifts [7] and label scarcity in real-world graph data impedes the ability of existing GNN models to adapt to new domains [8]. To addresses this challenge, Unsupervised Graph Domain Adaptation (UGDA) has become an important solution for transferring knowledge from a labeled source graph to an unlabeled target graph. This powerful paradigm has been widely studied, unlocking broader application for graph neural networks.

Despite a wide range of researches of UGDA have been developed [9, 10, 11, 12, 13, 14, 15, 16, 17, 18, 19, 20, 21, 22, 23, 24, 25, 26], the understanding of their capabilities and limitations is inadequate due to the following reasons:

- **Inadequate Evaluation of Domain Distribution Discrepancies.** The distribution shifts in node attributes, graph structures, and label proportions between graphs will significantly influence the adaptation performance and result in various adaptation scenarios [26, 27]. However, the types

---

\*Corresponding author

38th Conference on Neural Information Processing Systems (NeurIPS 2024) Track on Datasets and Benchmarks.

and magnitudes of distribution discrepancies among different domains have not been thoroughly evaluated and discussed, which makes it challenging to understand the robustness and efficacy of current methods.

- **Lack of Standard, Fair, and Comprehensive Comparisons.** The utilization of distinct datasets, varying data processing methodologies, and divergent data partitioning strategies among existing domain adaptation models results in incomparability across different findings [9, 28, 17]. Furthermore, they are mainly evaluated against limited baselines with constrained scenarios, such as social networks or citation networks, which lack validation of the model's capability in more diverse or complex applications.

- **Limited Investigation on GNN Inherent Transferability.** Despite the advancements made by existing UGDA algorithms, it is still unclear how data shift impose challenges on GNN and how to unleash the transferabililty power for GNN. Due to the non-IID nature of graph data, the aggregation architectures and aggregation scopes affect the underlying distribution of latent representations generated by GNN. When there exists significant structural difference between the source and target domains, such as variations in the degree [29, 30] or differences in subgraph patterns [23, 31], the information aggregation capability of GNN will be directly affected [32, 25, 31]. Thus, understanding the key components that affect adaptation in GNN will be crucial for enhancing GNN's transferabililty, which is still an open problem.

To fill these gaps, we revisit existing UGDA algorithms and conduct a comprehensive benchmark named GDABench. Specifically, GDABench includes 16 state-of-the-art UGDA models and diverse real-world graph datasets covering node attributes, graph structures, and label proportion shifts. Additionally, we also explore the limits of GNN transferability by combining 7 GNN variants with 2 domain alignment and 3 unsupervised graph learning techniques. Our work is the first to provide a rigorous empirical analysis of how various aggregation mechanisms influence alignments in domain adaptation task. Through comprehensive experiments, we observe that: (1) the performance of current UGDA models varies greatly across different datasets and adaptation scenarios; (2) it is crucial to develop tailored strategies to address graph structural shifts, especially when the distribution discrepancies are significant; (3) the GNN's transferability in UGDA heavily relies on two factors: aggregation scope and aggregation architecture, which are influenced by the severity of label shift and the level of graph heterophily, etc; (4) the inherent adaptability of GNNs is largely underestimated by existing methods, which motivates the exploration of a simple yet effective model that fully leverages the core property of GNN. More insights can be found in Section 5.

In summary, our main contributions are as follows:

- We introduce GDABench, the first comprehensive benchmark for unsupervised graph domain adaptation. It includes 16 recent state-of-the-art methods across various real-world datasets with diverse range of adaptation tasks.

- To explore the capability and limitations of exiting UGDA models, we systematically evaluate existing algorithms and investigate the underlying transferability for GNN. With these findings, we reveal a simple yet effective method that can even surpass existing UGDA algorithms.

- We develop an easy-to-use library PyGDA to alleviate the workload of researchers when conducting experiments. Furthermore, users can easily construct their own models or datasets with minimal effort.

The source codes of our benchmark are available at `https://github.com/pygda-team/pygda/tree/main/benchmark`, which provide unified APIs and adopt consistent data processing as well as data splitting approaches for fair comparisons.

## 2 Preliminaries and Related Work

### 2.1 Problem Definition

Consider a graph $\mathcal{G} = (\mathcal{V}, \mathcal{E})$ with $n$ nodes and $m$ edges. The node feature matrix, denoted by $\mathbf{X} = \{x_v | v \in \mathcal{V}\} \in \mathbb{R}^{n \times d}$, contains attribute vector for each node, where $d$ represents the dimensionality of the attributes. We denote adjacency matrix as $\mathbf{A} \in \mathbb{R}^{n \times n}$, where $\mathbf{A}_{i,j} = 1$ indicates the presence of an edge $e_{i,j} \in \mathcal{E}$ connecting node $v_i$ and $v_j$, and $\mathbf{A}_{i,j} = 0$ otherwise.

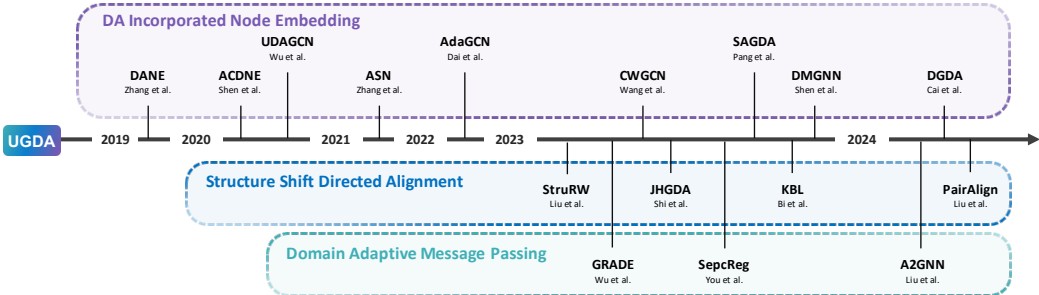

Figure 1: This timeline illustrates the diverse UGDA algorithms revisited in this paper. All of them are incorporated into our PyGDA library. More details are shown in Section 2 and Appendix C

$\mathbf{Y} \in \mathbb{R}^{N \times C}$ represents the node label matrix, where $C$ is the category of node labels. We focus on Unsupervised Graph Domain Adaptation (UGDA) on classification tasks. It is a non-trivial task due to the complex domain shift between source and target graphs. Formally, given a labeled source graph $\mathcal{G}_S = (\mathcal{V}_S, \mathcal{E}_S, \mathcal{Y}_S)$ and an unlabeled target graph $\mathcal{G}_T = (\mathcal{V}_T, \mathcal{E}_T)$ with the data shift that $\mathbb{P}_S(\mathcal{G}) \neq \mathbb{P}_T(\mathcal{G})$. The goal is to train a graph neural network model $h : \mathcal{G} \to \mathcal{Y}$ that utilizes the labeled source graph $\mathcal{G}^s$ and the unlabeled target graph $\mathcal{G}^t$ in order to make accurate label predictions for the target graph $\mathcal{G}^t$.

## 2.2 Related Work

**Classical Methods.** There are two classical categories to reduce the domain discrepancy [33, 34]: minimizing pre-defined probability discrepancy metrics [35, 36] and adversarial learning techniques [37, 38, 39, 40]. For the pre-defined probability discrepancy metric minimization models, node representations are initially derived from the encoder, after which domain-invariant representations are acquired by minimizing probability discrepancy distances, such as MMD [41], CMD [36], etc. Rather than directly minimizing domain discrepancies, some approaches integrate the encoder with a domain classifier that predicts the source domain of each representation. For example, DANN [37] facilitates domain invariant learning (DIRL) to distinguish between source and target samples in the latent space. Building upon this framework, WDGRL [42] replaces the domain classifier with a network that learns an approximate Wasserstein distance. These classical methods are designed for CV and NLP tasks, where samples are independently and identically distributed [43]. However, marginal alignment of node representations in non-graph DA research is insufficient for graph-structured data, of which the distribution shift becomes more complex due to the interconnection among different nodes.

**Specialized Methods for Graph Data.** To tackle the unique challenges of knowledge transfer in graphs, several approaches have been proposed [9, 11, 44, 13, 14, 15, 16, 17, 18, 19, 20, 21, 22, 23, 24, 25, 26]. The key idea behind existing work on node-level UGDA is to leverage node representations as intermediaries while minimizing domain shift through adversarial learning. This kind of methods incorporate classical DA methods with deep node embedding by designing a special encoder to learn transferable features. DANE [10] uses shared weight GCNs to get node representations and then handles distribution shift via least square generative adversarial network. ACDNE [11] employs two feature extractors to simultaneously maintain both attributed affinity and topological proximity through a deep network embedding module. Additionally, it integrates a domain classifier to enhance the label-discriminative nature of the node representations. UDAGCN [12] introduces a dual graph convolutional network enhanced by an attention mechanism, allowing it to leverage both local and global consistency for improved graph representation learning. ASN [44] distinguishes between domain-private and domain-shared information while integrating both local and global consistency to effectively capture network topology information. AdaGCN [13] utilizes GCNs to merge network topology with adversarial domain adaptation, effectively enhancing graph convolution processes. CWGCN [22] introduces a two-step correntropy-induced Wasserstein GCN. This method first eliminates noisy nodes from the source graph and subsequently learns the target GCN by extending the Wasserstein distance. SA-GDA [23] introduces a spectral augmentation module aimed at improving node representation learning by integrating spectral information from the target domain into the source domain. DMGNN [21] utilizes a GNN encoder equipped with

dual feature extractors to distinguish between ego-embedding learning and neighbor-embedding learning. Following this, a label propagation node classifier is applied to enhance the accuracy of label predictions. DGDA [24] approaches graph domain adaptation from a generative perspective, breaking down the generation process into three components: semantic latent variables, domain latent variables, and random latent variables.

Despite the progress made by the aforementioned UGDA models, such solutions still struggle to address the complex domain shift present in real-world graph data. The complex distribution shift between graphs usually combine node attribute shift, graph structure shift and label shift. Thus, to deal with the distinct effects of distribution shifts caused by graph structures, StruRW [17] investigates different types of distribution shifts of graph-structured data and reweights edges in the source graph to reduce the conditional shift of neighborhoods. Based on this work, PairAlign [26] addresses conditional structure shifts by recalibrating the influence of neighboring nodes using edge weights, while also modifying the classification loss through label weights to tackle label shifts. Except for reweighting, JHGDA [18] develops a hierarchical pooling model that extracts meaningful and adaptive structures at various levels. This model simultaneously minimizes both marginal and class conditional distribution shifts across each hierarchical layer. KBL [19] redefines the aggregation mechanism as a process of learning a knowledge-enhanced posterior distribution for target domains, facilitating knowledge transfer by linking informative samples across domains.

Instead of simply using GNN as a node embedding module, some methods devote effort to empirically and theoretically studying the role of GNN within domain adaptation [15, 16, 25]. This allows the UGDA method to be precisely customized based on the property of GNN, thereby enhancing their performance in graph domain adaptation. GRADE [15] introduces graph subtree discrepancy as a metric to measure the graph distribution shift by connecting GNNs with WL subtree kernel [45]. SpecReg [16] finds that the OT-based bound for graph is closely coupled with the Lipschitz constant of GNN and proposes spectral regularization to modulate the Lipschitz constant to restrict the target risk bound. A2GNN [25] further investigates the GNN's underlying generalization capability behind its architecture and finds propagation operation plays a pivotal role in the adaptation procedure. Based on this observation, A2GNN proposes a simple yet effective GNN framework, which stacks more propagation layers on target branch. The timeline of UGDA algorithms covered in GDABench is shown in Figure 1 and more detailed descriptions are provided in Appendix C.

## 3 Datasets

We have carefully selected 5 widely used public datasets that showcase a wide spectrum of distribution shifts across graphs for the node classification task. These include **Airport** which consists of three domains: Brazil (B), Europe (E) and USA (U); **Blog** that includes two domains: Blog1 (B1) and Blog2 (B2); **ArnetMiner** which encompasses three domains: DBLPv7 (D), Citationv1 (C) and ACMv9 (A); **Twitch** that includes six domains: Germany (DE), England (EN), Spain (ES), France (FR), Porutgal (PT) and Russia (RU); **MAG** that includes six domains like CN, US, JP, FR, RU, and DE. The selection criteria for these datasets are primarily based on three factors: the complexity of the distribution shift, the scale of the dataset, and the potential for downstream applications. From these datasets, we have included a comprehensive collection, comprising 74 distinct source-target adaptation pairs. Detailed information about each dataset is provided in Table 1, while the statistical methods used to quantify the types of domain shifts exhibited in the dataset are presented in Appendix B. The chosen datasets possess the following characteristics:

- **Wide range of distribution shift.** The graph distribution shifts between source and target domains can largely fall into three categories: feature shift, structure shift and label shift [32, 46, 26]. Our GDABench datasets encompass a diverse range of three distribution shifts across domains in varying degrees. The details are illustrated in Table 1 and more statistics are given in Appendix B. Specifically, the domains within Airport are dominated by structure shift, while domains in Blog, ArnetMiner, Twitch and MAG are affected by all kinds of shifts with different degrees.

- **Different scales with variant spans.** We categorize the size of the dataset by the average number of domain nodes. The small size (S) covers nodes below 5 thousand. The medium size (M) cover nodes below 10 thousand. The large size (L) covers nodes from 10 thousand to hundred thousand. Smaller datasets exhibit smaller differences in domain sizes, and vice versa. In the Blog and Airport, the size difference between the largest and smallest domains is less than 1000 nodes. In the

Table 1: Datasets used in GDABench reflecting a wide range of distribution shifts. '-' indicates no data shift exists. Circles (○, ◑ and ●) represent the degree of the corresponding shift between domains and Airport does not contain node features. The magnitude of shift is directly proportional to the filling area of the circle. The statistic manners and more details are provided in Appendix B.

| Dataset | Size | Feature Shift | Structure Shift | Label Shift | # Domains | # Labels | # Homo |
|---|---|---|---|---|---|---|---|
| Airport | S | - | ◑ | ○ | 3 | 4 | 0.52 |
| Blog | S | ○ | ○ | ○ | 2 | 6 | 0.40 |
| ArnetMiner | M | ◑ | ◑ | ○ | 3 | 5 | 0.83 |
| Twitch | M | ● | ● | ◑ | 6 | 2 | 0.59 |
| MAG | L | ● | ● | ● | 6 | 20 | 0.58 |

case of ArnetMiner and Twitch, this difference ranges between 1,000 and 10,000. For the MAG, this difference ranges between 10,000 to 100,000. This allows us to understand the impacts of varying domain size on the efficacy of adaptation task.

- **Various downstream application scenarios.** The GDABench datasets encompass multiple application scenarios, including citation relationships (ArnetMiner and MAG), social media interactions (Blog and Twitch) and routine connections (Airport). Specifically, in ArnetMiner and MAG, nodes represent academic papers and edges indicate citation relationship. ArnetMiner groups domains by publisher, while MAG by country. Blog and Twitch capture friendship within blog and gamer networks, respectively. Airport delineates routines connections, where airports serve as nodes connected by flight routes.

## 4 Compared Models

**Specialized UGDA Methods.** This group includes specifically designed algorithms for graph domain adaptation task. We compare 16 models including (1) nine methods incorporating classicial DA methods with deep node embedding: DANE [10], ACDNE [11], UDAGCN [12], ASN [44], AdaGCN [13], CWGCN [22], SAGDA [23], DMGNN [21] and DGDA [24]; (2) four methods tailored for graph structure shift: StruRW [17], JHGDA [18], KBL [19] and PairAlign [26]; and (3) three methods based on domain adaptive message passing: GRADE [15], SpecReg [16], and A2GNN [25].

**SimGDA: Vanilla DA with GNN Variants.** To understand the inherent transferability of GNN, we delve into its aggregation process by decoupling it into two key perspectives: *how to aggregate* and *what to aggregate*. For *how to aggregate*, we consider five types of aggregators, including sum aggregator [47], mean aggregator [48], aggregate with weighted neighbours (GAT) [49] and aggregate with discriminative neighbours (GIN) [50]. For *what to aggregate*, we consider three aspects in terms of the hop-count of neighbours: (1) GNN without neighbours, where graph structure is not considered (degenerating to MLP); (2) GNN with one-hop neighours; and (3) GNN with multi-hop neighours. To avoid the over-smooth problem, we also add residual connections to enhance its modeling power [51, 47]. For alignment, we consider two widely used models for domain-invariant feature learning from computer vision: domain distance metric MMD [35] and adversarial learning DANN [37]. Among them, MMD proposes to match the distribution in the latent space through maximum mean discrepancy [41], while DANN introduces an adversarial objective to distinguish source and target samples in the latent space. We use one-layer GCN [47] as a control and create six GNN variants by altering only one module each time. Then, we get 14 models by combining these variants with two vanilla DA methods, abbreviated these 14 models as SimGDA.

**SimGDA+: SimGDA with Unsupervised Techniques.** To further unlock the power of GNN for graph domain adaptation, we enhance SimGDA with different unsupervised graph learning techniques on unlabeled target graph, which allows the model to learn meaningful representations without relying on domain-specific labels. We implement three unsupervised techniques in

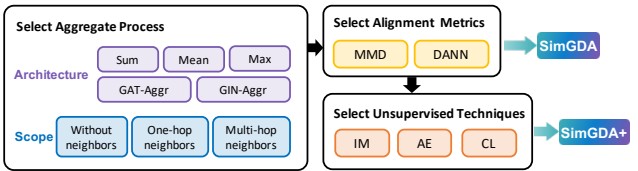

Figure 2: The combination process of SimGDA / SimGDA+.

Table 2: We compared Micro-F1 of each model on Airport, Blog, and ArnetMiner. Highlighted are the top **first**, second, and third results. Results for other tasks can be found in Appendix D.

| Models | Airport | | Blog | | ArnetMiner | |
|---|---|---|---|---|---|---|
| | E→U | U→E | B1→B2 | B2→B1 | C→A | D→A |
| DANE | $31.18_{\pm 3.26}$ | $33.75_{\pm 0.31}$ | $32.17_{\pm 3.20}$ | $32.77_{\pm 0.66}$ | $62.87_{\pm 1.98}$ | $59.19_{\pm 1.66}$ |
| ACDNE | $48.52_{\pm 1.17}$ | $45.03_{\pm 0.43}$ | $54.30_{\pm 1.42}$ | $56.93_{\pm 0.56}$ | $72.34_{\pm 0.39}$ | $65.37_{\pm 3.50}$ |
| UDAGCN | $41.62_{\pm 0.58}$ | $33.17_{\pm 0.12}$ | $27.58_{\pm 0.63}$ | $25.46_{\pm 5.96}$ | $70.24_{\pm 1.01}$ | $62.49_{\pm 1.16}$ |
| ASN | $46.58_{\pm 0.21}$ | $40.85_{\pm 0.54}$ | $53.91_{\pm 0.52}$ | $56.25_{\pm 0.52}$ | $71.70_{\pm 0.38}$ | $66.15_{\pm 1.02}$ |
| AdaGCN | $46.55_{\pm 0.42}$ | $49.62_{\pm 0.01}$ | $43.06_{\pm 3.03}$ | $36.58_{\pm 8.94}$ | $67.66_{\pm 0.36}$ | $60.47_{\pm 0.99}$ |
| DMGNN | $45.85_{\pm 1.03}$ | $27.82_{\pm 2.95}$ | $46.27_{\pm 2.40}$ | $44.96_{\pm 1.57}$ | $72.91_{\pm 0.44}$ | $70.68_{\pm 0.27}$ |
| CWGCN | $44.68_{\pm 0.42}$ | $40.69_{\pm 0.47}$ | $31.96_{\pm 3.47}$ | $33.46_{\pm 4.96}$ | $71.65_{\pm 0.21}$ | $68.21_{\pm 0.09}$ |
| SAGDA | $30.62_{\pm 5.57}$ | $35.92_{\pm 1.01}$ | $26.51_{\pm 11.12}$ | $26.91_{\pm 7.26}$ | $65.40_{\pm 4.38}$ | $64.60_{\pm 0.89}$ |
| DGDA | $43.45_{\pm 2.16}$ | $43.78_{\pm 2.90}$ | $22.10_{\pm 1.45}$ | $21.06_{\pm 2.07}$ | $52.20_{\pm 4.62}$ | $56.31_{\pm 2.01}$ |
| StruRW | $45.94_{\pm 0.69}$ | $36.09_{\pm 0.01}$ | $40.02_{\pm 0.37}$ | $42.10_{\pm 1.18}$ | $70.59_{\pm 0.15}$ | $64.15_{\pm 0.31}$ |
| KBL | $44.54_{\pm 0.73}$ | $32.08_{\pm 0.20}$ | $35.14_{\pm 3.97}$ | $34.90_{\pm 2.49}$ | $70.49_{\pm 0.26}$ | $63.34_{\pm 0.53}$ |
| JHGDA | $36.89_{\pm 0.25}$ | $40.85_{\pm 1.68}$ | $17.79_{\pm 2.12}$ | $23.16_{\pm 6.59}$ | $65.53_{\pm 0.94}$ | $60.80_{\pm 0.35}$ |
| PairAlign | $42.38_{\pm 0.77}$ | $36.84_{\pm 1.48}$ | $32.17_{\pm 10.88}$ | $41.16_{\pm 3.02}$ | $58.06_{\pm 2.62}$ | $56.68_{\pm 0.89}$ |
| GRADE | $49.36_{\pm 0.35}$ | $48.45_{\pm 1.56}$ | $38.64_{\pm 3.73}$ | $44.01_{\pm 4.51}$ | $69.16_{\pm 0.39}$ | $63.47_{\pm 1.10}$ |
| SpecReg | $37.59_{\pm 2.55}$ | $28.91_{\pm 8.77}$ | $28.27_{\pm 4.22}$ | $30.30_{\pm 1.35}$ | $68.90_{\pm 4.78}$ | $66.30_{\pm 4.28}$ |
| A2GNN | $50.64_{\pm 1.47}$ | $53.47_{\pm 0.24}$ | $22.58_{\pm 0.01}$ | $33.04_{\pm 4.12}$ | $76.15_{\pm 0.06}$ | $74.12_{\pm 0.18}$ |
| SimGDA | $55.29_{\pm 0.39}$ | $54.39_{\pm 0.90}$ | $53.35_{\pm 0.69}$ | $43.04_{\pm 0.86}$ | $70.80_{\pm 0.06}$ | $67.04_{\pm 0.15}$ |
| SimGDA+ | $58.11_{\pm 0.40}$ | $57.52_{\pm 0.38}$ | $57.04_{\pm 0.45}$ | $44.17_{\pm 0.02}$ | $73.18_{\pm 0.38}$ | $71.81_{\pm 2.44}$ |

an end-to-end manner: (1) Information Maximization (IM) [52, 53, 20]. Ideally, an accurate prediction for the target domain should exhibit individual certainty while maintaining global diversity. To accomplish this, we minimize the entropy for each individual sample while maximizing the entropy across classes. (2) Graph AutoEncoder (AE) [54, 55, 56]. Graph autoencoders encode nodes into a latent vector space and reconstruct the graph data from the encoded latent space. After obtaining target graph node representations, we employ a distinct decoder to reconstruct target graph structure. (3) Graph Contrastive Learning (CL) [57, 58, 59, 60]: Graph contrastive learning methods maximize mutual information between augmented instances of the same object (e.g., node).

As changing the graph structure may affect the estimation of structure shift, we take random attribute masking to create augmented instances for each target node. Following previous works [58, 60], we utilize a one-layer perceptron (MLP) as projection head to map augmented representations to the shared latent space and then calculate contrastive loss based on normalized temperature-scaled cross entropy loss (NT-Xent) [61]. As a result, we get 42 combined models, which integrates 14 SimGDA variants with 3 unsupervised techniques. We collectively refer to these combined models as **SimGDA+**, and the combination process is shown in Figure 2.

## 5 Experimental Results and Analyses

In this section, we study the experimental results of all the models. We first provide a comprehensive comparison of specialized UGDA methods across five datasets with diverse distribution shifts for node classification task. Following that, we perform a thorough analysis between different GNN variants to understand how data shift imposes challenges on GNNs. Finally, we present the performance of SimGDA variants, which shows the limit of GNNs' transferability. For more information about metrics, hyperparameters, search spaces, and other implementation details, please refer to Appendix D. We further extend our scope to include the graph-level classification task; please refer to the Appendix E for details.

### 5.1 Overall Comparisons

In this section, we try to elucidate the success of these UGDA algorithms through empirical evaluation across diverse datasets. In Table 2 and Table 3, we take a close look at the models' performance across 5 datasets on partial tasks utilizing Micro-F1 for Blog, Airport, and ArnetMiner, while employing

Table 3: We evaluated Macro-F1 on MAG and AUROC scores on Twitch. Highlighted are the top **first**, **second**, and **third** results. OOM indicates out of memory. Results for other tasks can be found in Appendix D

| Models | Twitch | | | MAG | | |
|---|---|---|---|---|---|---|
| | DE→ ES | EN → RU | DE → PT | FR → JP | JP → FR | JP → RU |
| DANE | $56.71_{\pm 0.57}$ | $53.47_{\pm 0.84}$ | $55.71_{\pm 1.70}$ | $16.16_{\pm 0.24}$ | $16.71_{\pm 1.45}$ | $12.61_{\pm 0.19}$ |
| ACDNE | $51.13_{\pm 0.34}$ | $50.79_{\pm 0.12}$ | $52.47_{\pm 1.83}$ | $20.12_{\pm 0.37}$ | $18.92_{\pm 1.08}$ | $13.91_{\pm 0.35}$ |
| UDAGCN | $56.48_{\pm 0.18}$ | $53.72_{\pm 0.42}$ | $54.22_{\pm 2.10}$ | $12.22_{\pm 0.31}$ | $11.62_{\pm 0.35}$ | $11.17_{\pm 0.29}$ |
| ASN | $53.57_{\pm 0.72}$ | $50.29_{\pm 0.15}$ | $55.03_{\pm 0.75}$ | $11.91_{\pm 1.45}$ | $12.04_{\pm 0.70}$ | $10.79_{\pm 0.24}$ |
| AdaGCN | $52.32_{\pm 0.76}$ | $51.99_{\pm 1.31}$ | $51.09_{\pm 0.61}$ | $16.21_{\pm 0.47}$ | $14.12_{\pm 0.46}$ | $13.05_{\pm 0.08}$ |
| DMGNN | $54.11_{\pm 0.09}$ | $50.42_{\pm 0.03}$ | $53.44_{\pm 0.04}$ | $12.01_{\pm 0.78}$ | $9.93_{\pm 0.40}$ | $10.28_{\pm 1.04}$ |
| CWGCN | $57.62_{\pm 0.64}$ | $52.90_{\pm 0.37}$ | $58.21_{\pm 0.57}$ | $11.01_{\pm 0.48}$ | $12.37_{\pm 0.54}$ | $12.38_{\pm 0.25}$ |
| SAGDA | $51.58_{\pm 0.09}$ | $51.03_{\pm 0.23}$ | $51.96_{\pm 0.70}$ | $16.28_{\pm 0.51}$ | $3.64_{\pm 5.07}$ | $11.40_{\pm 0.59}$ |
| DGDA | $54.43_{\pm 3.60}$ | $51.68_{\pm 1.08}$ | $54.29_{\pm 4.28}$ | OOM | OOM | OOM |
| StruRW | $59.60_{\pm 0.19}$ | $52.04_{\pm 0.36}$ | $58.74_{\pm 2.09}$ | $22.10_{\pm 0.40}$ | $12.89_{\pm 0.85}$ | $12.96_{\pm 0.43}$ |
| KBL | $58.33_{\pm 0.44}$ | $55.91_{\pm 0.16}$ | $51.66_{\pm 0.08}$ | $17.60_{\pm 0.39}$ | $6.12_{\pm 0.14}$ | $14.49_{\pm 0.30}$ |
| JHGDA | $62.25_{\pm 0.49}$ | $53.75_{\pm 0.15}$ | $61.88_{\pm 0.48}$ | $20.51_{\pm 0.20}$ | $20.46_{\pm 0.57}$ | $11.85_{\pm 0.37}$ |
| PairAlign | $50.78_{\pm 0.22}$ | $51.19_{\pm 0.20}$ | $52.03_{\pm 0.97}$ | $23.29_{\pm 0.49}$ | $23.72_{\pm 0.30}$ | $12.34_{\pm 0.31}$ |
| GRADE | $58.57_{\pm 0.42}$ | $53.55_{\pm 0.28}$ | $62.12_{\pm 0.17}$ | $11.93_{\pm 0.48}$ | $10.95_{\pm 0.55}$ | $9.35_{\pm 0.25}$ |
| SpecReg | $51.04_{\pm 0.33}$ | $50.17_{\pm 0.06}$ | $55.91_{\pm 0.59}$ | $19.45_{\pm 0.54}$ | $20.17_{\pm 1.35}$ | $15.82_{\pm 0.50}$ |
| A2GNN | $59.41_{\pm 0.34}$ | $52.01_{\pm 0.32}$ | $61.82_{\pm 0.77}$ | $26.20_{\pm 0.74}$ | $25.78_{\pm 0.25}$ | $16.94_{\pm 0.13}$ |
| SimGDA | $61.30_{\pm 0.32}$ | $53.11_{\pm 0.19}$ | $58.27_{\pm 0.16}$ | $18.59_{\pm 0.07}$ | $15.16_{\pm 0.11}$ | $13.27_{\pm 0.04}$ |
| SimGDA+ | $61.53_{\pm 0.08}$ | $53.82_{\pm 0.08}$ | $61.60_{\pm 0.11}$ | $21.94_{\pm 0.18}$ | $21.36_{\pm 0.09}$ | $15.64_{\pm 0.46}$ |

Macro-F1 for MAG and AUROC for Twitch due to their imbalanced labels. For comprehensive experimental results, please refer to Appendix D. Our key findings include:

***Observation 1: When facing significant shifts, it is important to design solutions tailored to mitigate structural discrepancies.*** Although several methods that incorporate classical DA approaches with deep node embedding have achieved impressive results on Airport, Blog and ArnetMiner datasets (e.g., ACDNE and DMGNN), they fail to obtain satisfied performance on datasets with significant shifts. These results underscore the limitations of marginal distribution alignment techniques in the presence of significant structural and label shifts in graph data. As shown in Table 3, methods tailored to address graph structure shifts show reasonable improvements over those that incorporate classical DA techniques with deep node embeddings. This suggests that mitigating the impact of graph structure shift on node representation learning under this scenario is crucial.

***Observation 2: Domain-adaptive message passing methods demonstrate superior and robust performance across a wide range of datasets and tasks.*** While methods that align marginal feature distributions and those designed for graph structure shifts can address datasets with mild and severe data shifts respectively, strategies specifically developed to leverage GNN properties demonstrate robustness and superior performance across diverse data shifts. As shown in Table 2 and Table 3, methods designed based on the inherent properties of GNN achieves the top-three best performance in 8 tasks out of 12 tasks. This finding suggests that leveraging the structural strengths of GCNs, combined with well-established domain adaptation principles, can result in an effective and efficient approach to addressing the challenges of domain variability in graph datasets. Such strategies represent a promising direction for future research and application in this field.

## 5.2 Understanding and unlocking the inherent power of GNN

Although many UGDA methods integrate traditional domain adaptation techniques, we observe their performance remains unsatisfactory and can even fall below that of SimGDA. This leads us to an intriguing question: do the intrinsic mechanisms of GNN play a more crucial role in enhancing transferability? To further investigate the question, we take one-layer GCN combined with vanilla DA as a baseline, and compare six variants: Max-Aggr, Mean-Aggr, GAT-Aggr, GIN-Aggr, with-no-neighbor, and with-multi-hop-neighbor; The results are shown in Figure 3. Moreover, we enhance

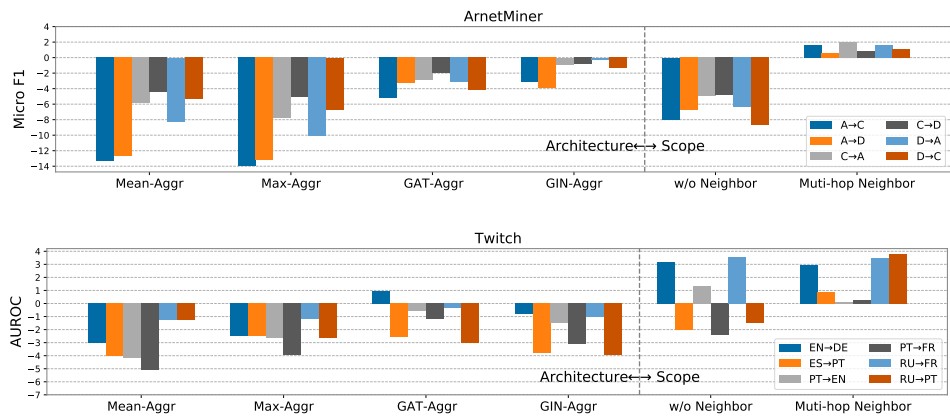

Figure 3: SimGDA: the compared performance of vanilla DA with 6 GNN variants.

above SimGDA with unsupervised graph learning techniques, referred to as SimGDA+, to explore the limit of GNNs in graph domain adaptation, which is shown in Table 2 and Table 3.

***Observation 3: SimGDA achieves competitive performance compared with UGDA methods.*** As shown in Table 2 and Table 3, SimGDA shows comparable results with state-of-the-art approaches across various datasets. Specifically, SimGDA exhibits better performance than structure shift tailored solutions in datasets with minor data shifts, i.e., Airport, Blog and ArnetMinver, and demonstrates advantages surpassing state-of-the-art methods in E → U and U → E. Similar trends are also observed in the Twitch and MAG dataset, where SimGDA outperforms most UGDA methods in six tasks. To further explore the impacts of data shifts on GNNs, we analyze the role of aggregation scope and mechanisms, which tackle diverse shifts through the graph structure.

***Observation 4: The benefit of multi-hop neighbors depends on the degree of label shift and graph heterophily.*** Across ArnetMiner, Blog, and Airport datasets, we observed a significant decline in performance for most tasks without neighbours, shown in Figure 3 and Appendix D. In contrast, on the Twitch and MAG datasets with heavy label shift, overlooking neighbor information generally benefits task performance. This suggests that the impact of structural information varies depending on the dataset with diverse degree of label shift.

Besides, we note a performance decline on the Airport and Blog datasets when including multi-hop neighbors. However, on the ArnetMiner, Twitch, and MAG datasets, incorporating multi-hop neighbors leads to performance improvement. The enhancement in performance in these cases can be attributed to the fact that heterophilous graphs exhibit a larger degree of conditional shift, and aggregation process may help to mitigate this situation by providing a more comprehensive view of the node's context.

***Observation 5: A source-unbiased discriminative aggregation mechanism is needed.*** As depicted in Figure 3, mean and max aggregators consistently exhibit lower performance across tasks compared to the sum aggregator used in GCN. This can be attributed to their inherent limitations in capturing discriminative structural information [50]. In contrast, the aggregator utilized in GCN can identify and distinguish between different structures by effectively incorporating degree-aware neighbors. Thus, the superiority of the GCN aggregator over mean and max emphasizes the necessity of a discriminative aggregation operator with highly expressive power.

To enhance the expressive power of aggregation operator, we evaluate aggregate mechanism used in GAT and GIN. Despite their stronger expressive capacities, their performance is generally ineffective across datasets such as ArnetMiner, Blog, Airport, and Twitch. This inefficiency can be attributed to the increased risk of model bias towards information from the source domain, stemming from the requirement of more parameters for learning. Consequently, source-biased discriminative aggregation mechanisms deteriorate the model's transfer capability.

***Observation 6: GNNs can serve as a powerful graph domain adaptor.*** With the optimal number of neighbor hops and aggregators, we further enhance SimGDA with unsupervised graph learning techniques and obtain SimGDA+. As can be seen in Table 2 and Table 3, SimGDA+ surpasses the

performance of most specially-designed methods for graph domain adaptation and even achieves the best performance in certain tasks. This superior performance under various types of shifts showcases the potential of GNNs as powerful domain adaptors. In summary, we contend that GNNs, when designed with appropriate aggregators, careful selection of neighbor hops, and the application of unsupervised graph learning techniques, can serve as effective and reliable graph domain adaptors.

## 5.3 Do LLMs help mitigate distribution shift in graphs?

Recently, Large Language Models (LLMs) [62] have demonstrated an impressive ability to understand and handle various text-related tasks. When dealing with text-attributed graph, an important question arises: does the distribution shift persist after leveraging LLMs as feature encoders?

To investigate this question, we utilize the prompts from TAPE [63], which allows us to assess the impact of LLM-based features on the model's performance. For datasets, we choose the widely used ogbn-arxiv dataset [64], which contains paper title and abstract text information. Each node represents an arXiv paper, with directed edges indicating citations from one paper to another. The objective is to predict the 40 subject areas of arXiv computer science papers, which are manually assigned by the authors and moderators of arXiv. We split the data into three disjoint domains based on the publication year of the papers, i.e. 1950-2016, 2016-2018, 2018-2020. The statistical details for each domain are shown in the Table 4.

Table 4: Statistics of the split ogbn-arxiv dataset.

| Domains | # Nodes | # Edges | # Homo | # Avg Degree |
|---------|---------|---------|--------|--------------|
| 1950-2016 | 69,499 | 237,163 | 0.6945 | 3.41 |
| 2016-2018 | 51,241 | 111,754 | 0.6886 | 2.18 |
| 2018-2020 | 48,603 | 60,403 | 0.7092 | 1.24 |

We explored two approaches to enhance the original node attributes:

- LLM enhanced text with word2vec embedding [65], which combines the title, abstract, and LLM-generated predictions and explanations into a single input. This composite text is then fed into word2vec. Then, the node features are obtained by averaging the embeddings of its combined input. We refer to it as arxiv-LLM-w2v.

- LLM enhanced text with BERT embedding [66], which feeds the same composite text into a pretrained DeBERTa. Then, the node features are obtained by sentence embedding. Note that we did not finetune the DeBERTa like TAPE [63], since we focus on unsupervised graph domain adaptation. We refer to it as arxiv-LLM-bert.

First, we use MMD to characterize the degree of feature shift among these three datasets, i.e., ogbn-arxiv, arxiv-LLM-w2v and arxiv-LLM-bert. We consider 3 adaptation tasks, and the results are shown in Table 5. As we can see, when word2vec is used to encode the LLM-enhanced text, the feature shift is reduced compared to the original node features. However, when BERT is used for encoding, the feature shift increases. This indicates that the choice of text encoding method significantly influences the degree of feature shift.

Table 5: Feature shift among domains in arxiv dataset by using MMD [41] as metric.

| Source | Target | ogbn-arxiv | arxiv-LLM-w2v | arxiv-LLM-bert |
|--------|--------|------------|---------------|----------------|
| 1950-2016 | 2016-2018 | 0.0405 | 0.0400 ↓ | 0.0427 ↑ |
| 1950-2016 | 2018-2020 | 0.0528 | 0.0535 ↑ | 0.0796 ↑ |
| 2016-2018 | 2018-2020 | 0.0148 | 0.0138 ↓ | 0.0149 ↑ |

Next, we choose 5 recent graph domain adaptation models to assess the impact of LLM-based features on the model's performance. Each experiment is repeated 3 times, and we report the average Micro-F1 score with standard deviation. As illustrated in Table 6, the performance of most baselines shows significant improvement with the arxiv-LLM-w2v dataset, whereas performance notably declines with the arxiv-LLM-bert dataset compared to the original ogbn-arxiv dataset. These results align with the MMD scores presented in the Table 5, which indicate that arxiv-LLM-bert exhibits a

Table 6: Micro-F1 score of 5 recent graph domain adaptation models with LLM-based features.

| Source | Target | Features | UDAGCN | AdaGCN | KBL | GRADE | A2GNN |
|--------|--------|----------|--------|--------|-----|-------|-------|
| 1950-2016 | 2016-2018 | ogbn-arxiv | 52.42±0.52 | 61.78±0.18 | 50.88±0.24 | 61.85±0.17 | 60.18±0.54 |
| | | arxiv-LLM-w2v | 43.55±0.90 | 66.23±0.14 | 64.90±0.12 | 65.41±0.31 | 63.77±0.36 |
| | | arxiv-LLM-bert | 34.81±0.29 | 35.66±1.20 | 34.64±0.85 | 35.04±0.86 | 39.14±2.00 |
| 1950-2016 | 2018-2020 | ogbn-arxiv | 48.41±0.64 | 56.74±0.34 | 47.53±1.15 | 57.19±0.26 | 58.89±0.28 |
| | | arxiv-LLM-w2v | 39.56±0.54 | 62.04±0.29 | 60.67±0.29 | 62.04±0.29 | 65.35±0.12 |
| | | arxiv-LLM-bert | 30.18±0.25 | 31.69±0.42 | 30.26±1.09 | 31.09±0.12 | 35.40±1.55 |
| 2016-2018 | 2018-2020 | ogbn-arxiv | 54.84±0.22 | 62.05±0.04 | 52.99±0.05 | 61.42±0.14 | 59.45±0.28 |
| | | arxiv-LLM-w2v | 47.20±0.99 | 68.42±0.04 | 66.77±0.13 | 66.84±0.16 | 65.83±0.40 |
| | | arxiv-LLM-bert | 39.20±2.53 | 39.93±1.90 | 37.27±2.06 | 34.14±1.54 | 43.62±1.87 |

larger distribution shift compared to the other datasets. This emphasizes the necessity of selecting an appropriate text encoder when utilizing LLM-enhanced text for graph domain adaptation. An effective choice of text encoder can greatly impact the performance and mitigate the distribution shifts in text-attributed graph domain adaptation tasks.

# 6 Conclusion

In this paper, we introduce GDABench, the first comprehensive benchmark for unsupervised graph domain adaptation. Our evaluation encompasses 16 well-known models across various real-world datasets exhibiting diverse data distribution shifts. Furthermore, we also designed 6 GNN variants to investigate the inherent transferability of GNNs, enhancing them with 3 unsupervised techniques to explore their potential limits. Our empirical results shows that (1) the performance of current UGDA models varies significantly across different datasets and adaptation scenarios; (2) tailored strategies are essential for addressing and mitigating graph structural shifts, particularly when distribution discrepancies are substantial. (3) the transferability of GNNs in UGDA is heavily dependent on aggregation scope and architecture, influenced by factors such as label shift severity and graph heterophily. We have provided unified APIs and adopted consistent data processing as well as data splitting approaches for fair comparisons. In the future, we plan to extend GDABench to include broader scenarios [67, 68], more cutting-edge models and more complex types of datasets. We hope our benchmark and findings will promote realistic and rigorous evaluations, inspiring new advances in graph domain adaptation.

**Border Impacts and Limitations.** Our benchmark fosters innovation and advances research in graph domain adaptation by providing a standardized evaluation platform, leading to the development of more effective algorithms. This standardization helps researchers compare methods more fairly, driving progress and collaboration within the field. However, benchmark datasets may introduce limitations that could impact the generalization of findings to real-world scenarios. This risk includes the potential for unrealistic performance expectations if the benchmark does not adequately represent the diversity and complexity of real-world data. We plan to enhance GDABench by including more settings such as source-free and open-set scenarios. This expansion will help to cover a wider range of domain adaptation challenges, thereby fostering the development of algorithms that are not only more robust but also versatile enough to navigate the complexities of diverse and dynamic real-world scenarios. This trajectory in research will be pivotal in advancing the capabilities of domain adaptation techniques, ensuring their applicability and efficacy across various domains and evolving data landscapes.

# Acknowledgments and Disclosure of Funding

This work is supported by the National Natural Science Foundation of China (62372408, 62106221), Zhejiang Provincial Natural Science Foundation of China (Grant No: LTGG23F030005), Ningbo Natural Science Foundation (Grant No: 2022J183). The research of Zhen Zhang and Bingsheng He is supported by the National Research Foundation, Singapore under its Industry Alignment Fund – Pre-positioning (IAF-PP) Funding Initiative. Any opinions, findings and conclusions or recommendations expressed in this material are those of the author(s) and do not reflect the views of National Research Foundation, Singapore.

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

# Appendix

## A    Distribution Shift in Graph-Structured Data

Distribution shift appears when the joint distribution differs between source domain and target domain [7, 69]. Assuming that the relationship between the input and class variables is unchanged, there are two kinds of distribution shift, i.e., covariate shift and label shift (prior probability shift) [70].

### A.1    Covariate Shift

Covariate shift [71] refers to changes in the distribution of the input variables, which can be defined formally as follows:

***Definition 1*** (Covariate Shift). Covariate shift appears when $\mathbb{P}_S(G) \neq \mathbb{P}_T(G)$ with the assumption of $\mathbb{P}_S(Y|G) = \mathbb{P}_T(Y|G)$, where $\mathbb{P}_S$ and $\mathbb{P}_T$ are the probability distributions of the source and target domains, respectively.

To deal with covariate shift, it is essential to align $\mathbb{P}_S(Y|H)$ and $\mathbb{P}_T(Y|H)$, where $H$ is the representation after data attributes passing through the encoder. However, in graph-structured data, node representation is not only affected by the data attributes but also graph structure. Thus, covariate shift in graph data can be decoupled as feature shift and structure shift [26].

***Definition 2*** (Feature Shift). Given the joint distribution of the node attributes and node labels $\mathbb{P}_T(X, Y)$, the feature shift is then defined as $\mathbb{P}_S(X, Y) \neq \mathbb{P}_T(X, Y)$ with the assumption of $\mathbb{P}_S(Y|G) = \mathbb{P}_T(Y|G)$.

***Definition 3*** (Structure Shift). Given the joint distribution of the adjacency matrix and node labels $\mathbb{P}_T(A, Y)$, the structure shift is then defined as $\mathbb{P}_S(A, Y) \neq \mathbb{P}_T(A, Y)$ with the assumption of $\mathbb{P}_S(Y|G) = \mathbb{P}_T(Y|G)$.

### A.2    Label Shift

Label shift refers to changes in the distribution of the class variable $Y$. It also appears with different names in the literature and the definitions have slight differences between them.

***Definition 4*** (Label Shift). Label shift occurs when the distribution of labels changes across two domains, which is defined as $\mathbb{P}_S(Y) \neq \mathbb{P}_T(Y)$ where $\mathbb{P}_S(G|Y) = \mathbb{P}_T(G|Y)$.

In all, structure shift is unique to graph data due to the non-IID nature caused by node interconnections. Moreover, the learning of node representations implemented by the GNN will mix the feature shift, sutructure shift and label shift [32].

## B    Detailed Description of Datasets

In this section, we provide additional details about the datasets used in our benchmark.

### B.1    Dataset Description

- **Airport**[2]: The Airport datasets consist of three separate collections corresponding to Brazil (B), Europe (E), and the USA (U). In these datasets, nodes represent airports and edges denote flight connections between them. The labels categorize airports by activity levels, measured in terms of flights or passenger numbers.

- **Blog**[3]: Blog1 and Blog2 are disjoint social networks derived from the BlogCatalog dataset. In these networks, nodes correspond to bloggers, and edges reflect friendships among them. The attributes for each node consist of keywords from the blogger's self-description, and each node is assigned a label denoting its group affiliation. Given that both Blog1 and Blog2 originate from the same underlying network, their data distributions are nearly identical.

---

[2]https://github.com/GentleZhu/EGI/tree/main/data
[3]https://github.com/shenxiaocam/ACDNE/tree/master/ACDNE_codes/input

Table 7: Dataset Statistics.

| Dataset | # Domains | # Nodes | # Edges | # Homo | # Avg Degree | # Feat Dims | # Labels |
|---|---|---|---|---|---|---|---|
| Airport | USA (U) | 1,190 | 27,198 | 0.6978 | 22.86 | | |
| | BRAZIL (B) | 131 | 2,148 | 0.4683 | 16.40 | 241 | 4 |
| | EUROPE (E) | 399 | 11,990 | 0.4048 | 30.05 | | |
| Blog | Blog1 (B1) | 2,300 | 66,942 | 0.3991 | 29.11 | | |
| | Blog2 (B2) | 2,896 | 107,672 | 0.4002 | 37.18 | 8,189 | 6 |
| ArnetMiner | DBLPv7 (D) | 5,484 | 16,234 | 0.8198 | 2.96 | | |
| | ACMv9 (A) | 9,360 | 31,112 | 0.7998 | 3.32 | 6,775 | 5 |
| | Citationv1 (C) | 8,935 | 30,196 | 0.8598 | 3.38 | | |
| Twitch | England (EN) | 7,126 | 35,324 | 0.5560 | 4.96 | | |
| | Germany (DE) | 9,498 | 153,138 | 0.6322 | 16.14 | | |
| | France (FR) | 6,549 | 112,666 | 0.5595 | 17.20 | 3,170 | 2 |
| | Russia (RU) | 4,385 | 37,304 | 0.6176 | 8.51 | | |
| | Spain (ES) | 4,648 | 59,382 | 0.5800 | 12.78 | | |
| | Porutgal (PT) | 1,912 | 31,299 | 0.5708 | 16.40 | | |
| MAG | China (CN) | 101,952 | 285,991 | 0.5307 | 2.81 | | |
| | Germany (DE) | 43,032 | 127,704 | 0.5311 | 2.97 | | |
| | France (FR) | 29,262 | 79,182 | 0.5732 | 2.71 | 128 | 20 |
| | Janpan (JP) | 37,498 | 91,412 | 0.5645 | 2.44 | | |
| | Russia (RU) | 32,833 | 68,294 | 0.7682 | 2.08 | | |
| | USA (US) | 132,558 | 702,482 | 0.5174 | 5.30 | | |

- **ArnetMiner**[4]: These datasets comprise paper citation networks sourced from three distinct origins as provided by ArnetMiner [72]: "ACMv9" (A), "Citationv1" (C), and "DBLPv7" (D). Each dataset's nodes symbolize papers, while edges reflect their citation relationships. Specifically, "ACMv9" (A) includes papers from ACM spanning 2000 to 2010, "Citationv1" (C) consists of papers from the Microsoft Academic Graph up to 2008, and "DBLPv7" (D) contains papers from DBLP collected between 2004 and 2008. The aim is to categorize all papers into five specific research areas: Databases, Artificial Intelligence, Computer Vision, Information Security, and Networking.

- **Twitch**[5]: Twitch gamer networks from six regions—Germany (DE), England (EN), Spain (ES), France (FR), Portugal (PT), and Russia (RU)—comprise nodes representing users and connections that signify friendships among them. Node features include data on users' preferred games, geographical location, and streaming habits, among others. Users within these networks are categorized into two groups based on their use of explicit language.

- **MAG**[6]: The MAG dataset, a subset of the Microsoft Academic Graph, is a heterogeneous network featuring four distinct types of entities: papers (736,389 nodes), authors (1,134,649 nodes), institutions (8,740 nodes), and fields of study (59,965 nodes). It includes four varieties of directed relationships linking pairs of entity types: an author's affiliation with an institution, an author's authorship of a paper, paper citations, and papers' association with fields of study. Each paper node is enriched with a 128-dimensional word2vec feature vector, while the other entities lack input node features. The primary task within this dataset involves predicting the publication venue (conference or journal) for each paper, leveraging information about its content, cited references, authors, and the affiliations of these authors. Following PairAlign [26], we split the original dataset into six countries.

## B.2 Shift Statistics of Datasets

According to dataset statistics, shown in Table 7 and Figure 4, we measure the degree of domain shift exhibited in the datasets for each tasks using statistical methods. We use MMD [41], CSS [26], Kullback-Leibler Divergence to characterize the degree of feature shift, structure shift and label shift. The results of each tasks is shown in Table 13. We take the average results of all tasks as the shift statistics for the datasets, shown in Table 8. The 74 tasks compiled by the five carefully selected datasets can cover all combinations of domain shift scenarios.

---

[4]https://github.com/yuntaodu/ASN/tree/main/data
[5]http://snap.stanford.edu/data/twitch-social-networks.html
[6]https://zenodo.org/records/10681285

Table 8: Domain shifts statistics of GDABench datasets.

| Dataset | Size | Feature Shift | Structure Shift | Label Shift | Domain Num |
|---------|------|---------------|-----------------|-------------|------------|
| Blog | S | 0.0132 | 0.0802 | 0.2532 | 2 |
| Airport | S | 0 | 0.2769 | 0.0351 | 3 |
| ArnetMiner | M | 0.0241 | 0.2074 | 1.1519 | 3 |
| Twitch | M | 0.0468 | 0.3264 | 8.6949 | 6 |
| MAG | L | 0.0499 | 0.3960 | 25.7725 | 6 |

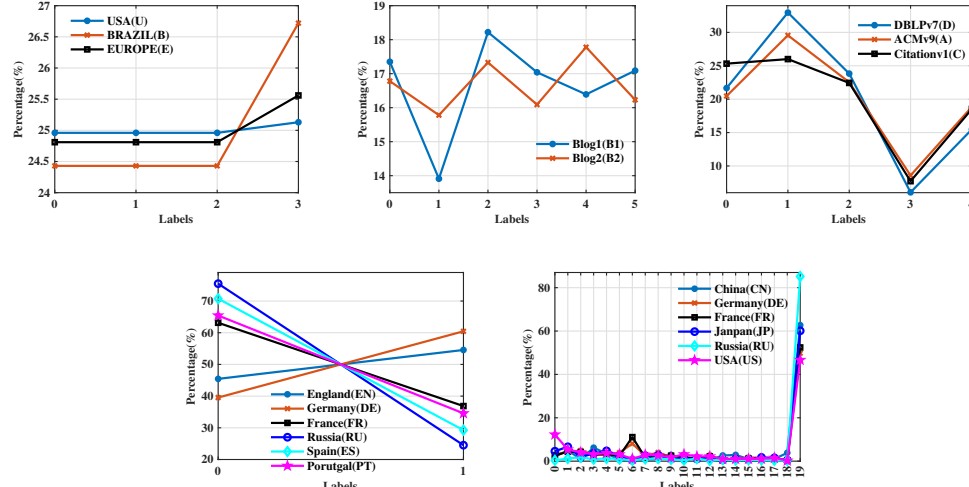

Figure 4: Label distribution of GDABench datasets.

- **Feature shift determined**: Tasks ES → PT, PT→ES, EN→DE, ED→EN, FR→ES, FR→PT, ES→FR, PT→FR, RU→ES, ES→RU, RU→PT, PT→RU, RU→FR and FR→RU in Twitch. Tasks JP→US, US→JP, JP→CN and CN→JP in MAG.

- **Sturcture shift determined**: Tasks E→B, B→E, U→B, B→U, U→E and E→U in Airport. Tasks GP→DE and US→DE in MAG. Tasks FR→DE, RU→EN, RU→DE, ES→DE, EN→RU, DE→RU, DE→ES and DE→PT in Twitch.

- **Lable shift determined**: Task FR→DE in MAG.

- **Determined by both feature and structure shift**: Tasks D→A, D→C, A→D and C→D in ArmetMiner. Tasks FR→EN, EN→FR, PT→EN, EN→PT, DE→FR, FR→DE, PT→DE and EN→ES in Twitch. Tasks JP→FR, RU→PT, RU→CN and DE→JP in MAG.

- **Determined by both feature and label shift**: Tasks EN→US, US→EN in MAG.

- **Determined by both structure and label shift**: Tasks DE→US, FR→US, US→FR, FR→RU in MAG.

- **All shifts effects**: Tasks B1→B2 and B2→B1 in Blog. Tasks A→C and C→A in ArnetMiner. Tasks DE→FR, CN→FR, JP→RU, RU→FR, CN→RU, FR→JP, RU→DE, CN→DE, RU→US, DE→CN, FR→CN, DE→RU and US→RU in MAG.

## C  GDA Baselines

MLP, GCN [47], GAT [49], and GIN [50] are classical GNN models. We directly adopt the implementation from Pytorch Geometric. The publicly available implementations of baselines can be found at the following URLs:

- **DANE** [10] uses shared weight GCNs to get node representations and then handles distribution shift via least square generative adversarial network. The source code is available at https://github.com/Jerry2398/DANE-Simple-implementation.

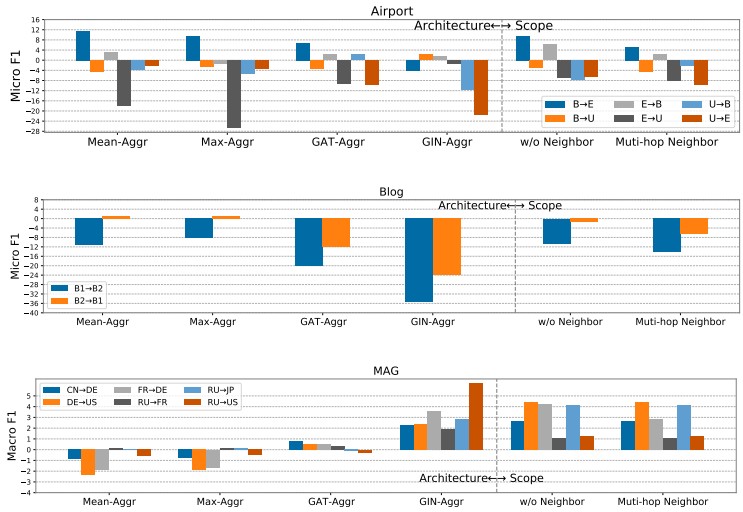

Figure 5: The compared performance of vanilla DA with 6 GNN variants.

- **ACDNE** [11] utilizes two feature extractors to jointly preserve attributed affinity and topological proximities as deep network embedding module and incorporates a domain classifier to make node representations label-discriminative. The source code is available at `https://github.com/shenxiaocam/ACDNE`.

- **UDAGCN** [12] develops a dual graph convolutional network with attention mechanism to jointly exploit local and global consistency for effective graph representation learning. The source code is available at `https://github.com/GRAND-Lab/UDAGCN`.

- **ASN** [44] separates domain-private and domain-shared information and combines local and global consistency to capture network topology information. The source code is available at `https://github.com/yuntaodu/ASN`.

- **AdaGCN** [13] leverages GCN to integrate network topology and combines adversarial domain adaptation with graph convolution. The source code is available at `https://github.com/daiquanyu/AdaGCN_TKDE`.

- **StruRW** [17] investigates different types of distribution shifts of graph-structured data and reweights edges in the source graph to reduces the conditional shift of neighborhoods. The source code is available at `https://github.com/Graph-COM/StruRW`.

- **GRADE** [15] introduces graph subtree discrepancy as a metric to measure the graph distribution shift by connecting GNNs with WL subtree kernel [45]. The source code is available at `https://github.com/jwu4sml/GRADE`.

- **SpceReg** [16] finds the OT-based bound for graph is closely coupled with the Lipschitz constant of GNN and proposes spectral regularization to modulate the Lipschitz constant to restrict the target risk bound. The source code is available at `https://github.com/Shen-Lab/GDA-SpecReg`.

- **A2GNN** [25] further investigates the GNN's underlying generalization capability behind its architecture and finds propagation operation plays a pivotal role. Based on this observation, A2GNN proposes a simple yet effective GNN which stacks more propagation layers on target branch. The source code is available at `https://github.com/Meihan-Liu/24AAAI-A2GNN`.

- **JHGDA** [18] designs a hierarchical pooling model to extract meaningful and adaptive hierarchical structures and jointly minimizes marginal and class conditional distribution shifts on each hierarchical level. The source code is available at `https://github.com/Skyorca/JHGDA`.

- **KBL** [19] redefines the aggregate mechanism as learning a knowledge-enhanced posterior distribution for target domains, which learns the scope of knowledge transfer by connecting knowledgeable samples between domains. The source code is available at `https://github.com/wendongbi/Bridged-GNN`.

- **DMGNN** [21] employs a GNN encoder with dual feature extractors to separate ego-embedding learning from neighbor-embedding learning and then a label propagation node classifier is employed

Table 9: We evaluated the Micro-F1 score on Airport and ArnetMiner.

| Models | Airport | | | | ArnetMiner | | | |
|---|---|---|---|---|---|---|---|---|
| | B → E | B → U | E → B | U → B | A → C | A → D | C → D | D → C |
| DANE | 33.00 | 41.23 | 41.98 | 39.44 | 64.40 | 62.52 | 66.13 | 71.30 |
| ACDNE | 46.45 | 56.30 | 55.73 | 64.12 | 79.07 | 74.27 | 75.47 | 79.06 |
| UDAGCN | 43.78 | 35.49 | 45.29 | 37.91 | 78.21 | 72.98 | 76.14 | 72.15 |
| ASN | 53.05 | 46.58 | 62.34 | 49.36 | 78.68 | 72.02 | 75.57 | 77.58 |
| AdaGCN | 50.63 | 43.47 | 60.56 | 61.32 | 73.87 | 66.91 | 72.56 | 71.20 |
| DMGNN | 33.92 | 29.92 | 35.37 | 34.10 | 81.59 | 76.62 | 76.77 | 80.65 |
| CWGCN | 46.37 | 46.58 | 58.78 | 44.27 | 80.00 | 74.29 | 76.23 | 76.95 |
| SAGDA | 35.51 | 37.76 | 47.33 | 48.35 | 77.5 | 70.56 | 74.03 | 59.49 |
| DGDA | 49.71 | 33.56 | 44.02 | 49.36 | 64.48 | 57.85 | 63.29 | 57.98 |
| StruRW | 56.06 | 43.36 | 65.65 | 61.32 | 77.24 | 67.51 | 74.37 | 73.96 |
| KBL | 45.28 | 45.52 | 51.40 | 33.84 | 77.71 | 69.16 | 74.48 | 74.62 |
| JHGDA | 48.87 | 40.59 | 65.14 | 43.51 | 73.74 | 69.13 | 71.71 | 71.59 |
| PairAlign | 39.93 | 42.18 | 51.91 | 54.96 | 68.29 | 61.80 | 62.89 | 63.28 |
| GRADE | 52.88 | 49.22 | 75.83 | 49.62 | 74.09 | 69.18 | 72.57 | 73.12 |
| SpecReg | 48.87 | 44.20 | 63.36 | 40.97 | 80.81 | 73.16 | 74.60 | 71.96 |
| A2GNN | 53.13 | 54.54 | 62.34 | 59.29 | 82.64 | 77.43 | 78.13 | 81.54 |
| SimGDA | 55.64 | 53.11 | 60.31 | 62.60 | 79.91 | 75.16 | 75.95 | 77.31 |
| SimGDA+ | 58.40 | 57.56 | 72.14 | 67.18 | 82.97 | 76.60 | 77.50 | 82.09 |

to refine label prediction. The source code is available at `https://github.com/shenxiaocam/DM_GNN`.

- **CWGCN** [22] puts forward a two-step correntropy-induced Wasserstein GCN, which first suppresses the noisy nodes in the source graph and then learns the target GCN based on extending the Wasserstein distance. The source code is available at `https://github.com/CocoLab-2022/CW-GCN`.

- **SAGDA** [23] proposes a spectral augmentation module to enhance the node representation learning, which combines the target domain spectral information within the source domain. Since the authors did not release the source code, we try our best to reproduce their results.

- **DGDA** [24] addresses graph domain adaptation in a generative view, which disentangles the generation process into the semantic latent variables, the domain latent variables, and the random latent variables. The source code is available at `https://github.com/rynewu224/GraphDA`.

- **PairAlign** [26] not only uses edge weights to recalibrate the influence among neighboring nodes to handle conditional structure shift but also adjusts the classification loss with label weights to handle label shift. The source code is available at `https://github.com/Graph-COM/Pair-Align`.

# D    Other Information in GDABench

We implement our GDABench library in PyTorch [73] and provide an infrastructure to run all the experiments to generate corresponding results. We have stored all models and logged all hyperparameters to facilitate reproducibility. Our framework can be easily extended to include new algorithms.

## D.1    Metrics

Following previous works [44, 12], we present the experiment performance on target domain. We select Area Under the Receiver Operating Characteristic Curve (AUROC) for Twitch, Micro-F1 for Airport, Blog and ArnetMiner and Macro-F1 for MAG.

- **AUROC** measures how well a model can distinguish between positive and negative classes by looking at the area under the ROC curve. This curve shows the true positive rate versus the false positive rate at various thresholds. An AUROC score of 1 means perfect distinction, while a score of 0.5 indicates the model does no better than guessing randomly.

- **Macro-F1** calculates the F1 score for each category independently and then taking the average of these scores. This method treats all categories equally, regardless of their frequency in the dataset.

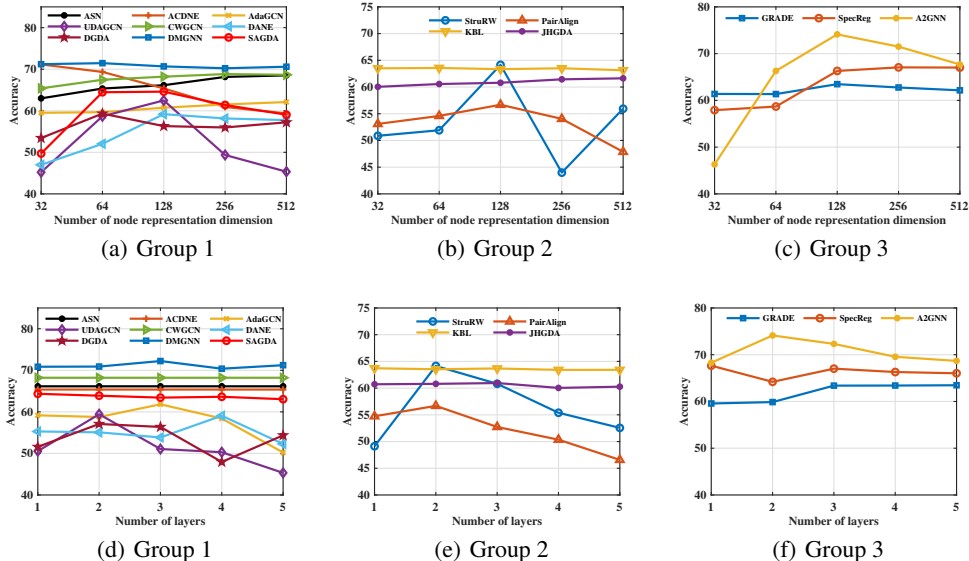

Figure 6: The impact of node representation dimension and the number of layers on ArnetMiner dataset (D→A). We classify all baselines into three groups: DA incorporated node embedding methods (Group 1), structure shift directed alignment (Group 2) and domain adaptive message passing (Group 3). The first row illustrates the impact of node representation dimension, while the second row presents the effect of the number of layers.

It is particularly useful when you want to understand the model's performance across smaller or less frequent categories, ensuring that performance on rare categories has as much weight as performance on more common ones.

- **Micro-F1** computes the average F1 score. This is achieved by summing up the true positives, false positives, and false negatives of the model across all categories and then calculating the F1 score using these totals. As a result, Micro-F1 gives a higher weight to the performance on more frequent categories, making it a useful metric when you're interested in understanding how the model performs on the majority of cases or the overall dataset.

## D.2 Additional Experimental Details

- **Hardware Specifications.** The experiments were conducted on a Linux server equipped with an Intel(R) Xeon(R) Platinum 8163 CPU operating at 2.50GHz, running Ubuntu 18.04.5 LTS. For GPU resources, we utilized a single NVIDIA Tesla V100 graphics card with 32GB of memory. The Python libraries employed for implementing our experiments include Python 3.8, PyTorch 1.13.1, PyTorch Geometric 2.4.0, PyTorch Sparse 0.6.15, and PyTorch Scatter 2.1.0.

- **Hyperparameter Settings.** To control the effect of hyperparameter selection and ensure fairness, we standardize the evaluation process with hyperparameter tuning. We utilize grid search to form the predefined search space for each models. We use all the source nodes and target nodes for model training. The experiments are repeated three times, and we report the mean performance. Table 14 provides a comprehensive list of all hyperparameters used in our grid search.

- **More Experimental Results.** In accordance with Table 2 and 3, we provide the performance for all tasks of each model in Table 9, 11 and 12. In accordance with Figure 3, we provide the compared performance of vanilla DA with 6 GNN variants in Figure 5.

- **Exploration of Hyperparameter Impact.** We investigate how various hyperparameters in common modules influence the performance of different UGDA methods on ArnetMiner dataset (task D → A). We focus on two key aspects: the number of GNN layers and the representation dimensions. Results are shown in Figure 6.

- **Running Time and Memory Consumption.** We also demonstrate the running time and memory consumption of each model on S/M/L datasets respectively. For time consumption, we evaluate the

efficiency of baselines by measuring the time it takes to converge. As shown in Figure 7, we can observe that some algorithms (e.g. A2GNN) can achieve relatively good performance with less complexity.

### D.3 The PyGDA Library

PyGDA is a Python library for Graph Domain Adaptation built upon PyTorch and PyG to easily train graph domain adaptation models in a sklearn style. PyGDA includes 15+ graph domain adaptation models. See examples with PyGDA below!

Graph Domain Adaptation Using PyGDA with 5 Lines of Code

```python
from pygda.models import A2GNN

# choose a graph domain adaptation model
model = A2GNN(in_dim=num_features, hid_dim=args.nhid,
num_classes=num_classes, device=args.device)

# train the model
model.fit(source_data, target_data)

# evaluate the performance
logits, labels = model.predict(target_data)
```

PyGDA is featured for:

- Consistent APIs and comprehensive documentation.
- Cover 15+ graph domain adaptation models.
- Scalable architecture that efficiently handles large graph datasets through mini-batching and sampling techniques.
- Seamlessly integrated data processing with PyG, ensuring full compatibility with PyG data structures.

## E   Experiments on Graph Classification

To expand our research scope, we take graph-level shifts into consideration and add a pooling layer to evaluate capabilities of baselines in graph-level domain adaptation. We employ three TUdatasets: Proteins, Mutagenicity, and Frankenstein, partitioning each dataset into 2 equally sized disjoint groups based on density shifts. Detailed statistics are shown in Table 10.

Table 10: Statistics of graph-level datasets in GDABench.

| Dataset | # Nodes | # Edges | # Feature | # Class | Num of graphs |
|---------|---------|---------|-----------|---------|---------------|
| Proteins | 39.06 | 72.82 | 4 | 2 | 1,113 |
| Mutagenicity | 30.32 | 30.77 | 14 | 2 | 4,337 |
| Frankenstein | 16.90 | 17.88 | 780 | 2 | 4,337 |

The results are detailed in Table 15 and Table 16. Among the methods, GRADE and A2GNN are domain adaptive message passing methods and the remaining are DA incorporated node embedding methods. Key observations are as follows:

*DA incorporated node embedding methods shows task-inconsisteny across node and graph-level tasks.* For example, DANE performs averagely in node-level tasks, but its performance improves significantly in graph-level tasks. This disparity highlights a challenge in predicting the performance of unsupervised graph domain adaptation (UGDA) models in real-world applications. The inconsistency suggests that models optimized for node-level tasks may not generalize well to graph-level tasks and

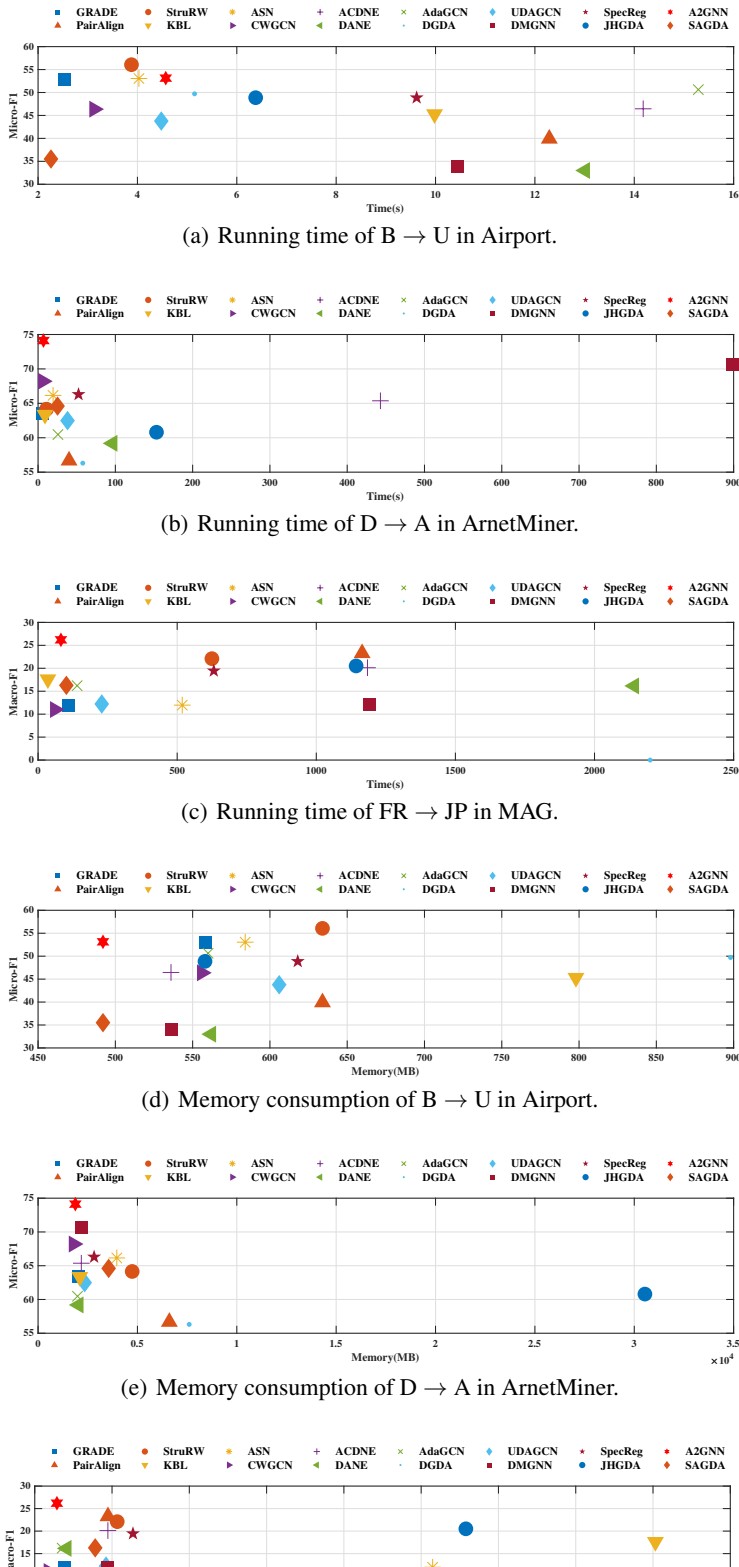

(a) Running time of B → U in Airport.

(b) Running time of D → A in ArnetMiner.

(c) Running time of FR → JP in MAG.

(d) Memory consumption of B → U in Airport.

(e) Memory consumption of D → A in ArnetMiner.

(f) Memory consumption of FR → JP in MAG.

Figure 7: Running time and memory consumption of baselines.

Table 11: We evaluated the AUROC score on Twitch.

| Models | DE → EN | DE → FR | EN → DE | EN → ES | EN → FR | EN → PT | EN → RU | ES → DE | ES → EN |
|---|---|---|---|---|---|---|---|---|---|
| DANE | 56.95 | 59.67 | 67.10 | 60.67 | 62.29 | 62.11 | 53.64 | 64.37 | 58.96 |
| ACDNE | 51.29 | 53.61 | 52.78 | 52.83 | 53.45 | 52.52 | 50.65 | 56.21 | 55.96 |
| UDAGCN | 54.70 | 54.51 | 56.95 | 53.80 | 54.64 | 51.55 | 50.76 | 55.35 | 56.58 |
| ASN | 51.00 | 51.79 | 55.17 | 53.95 | 50.64 | 54.56 | 51.01 | 56.96 | 54.64 |
| AdaGCN | 51.64 | 57.33 | 56.28 | 53.05 | 52.37 | 56.23 | 50.52 | 59.02 | 56.34 |
| DMGNN | 53.08 | 51.44 | 53.94 | 53.70 | 51.09 | 52.09 | 50.46 | 51.90 | 52.41 |
| CWGCN | 54.85 | 52.81 | 60.47 | 56.97 | 51.14 | 60.09 | 52.52 | 65.00 | 56.66 |
| SAGDA | 54.39 | 53.58 | 54.82 | 51.99 | 54.75 | 52.57 | 50.95 | 54.48 | 53.34 |
| DGDA | 56.67 | 52.63 | 62.77 | 60.47 | 56.37 | 57.75 | 52.25 | 62.64 | 54.00 |
| StruRW | 51.31 | 56.26 | 51.87 | 58.65 | 51.68 | 59.41 | 52.92 | 58.60 | 53.90 |
| KBL | 52.01 | 63.93 | 66.43 | 60.34 | 61.01 | 52.48 | 55.42 | 68.86 | 61.87 |
| JHGDA | 57.50 | 59.04 | 62.43 | 57.67 | 56.96 | 56.85 | 51.16 | 52.22 | 53.36 |
| PairAlign | 51.05 | 52.12 | 52.13 | 51.93 | 51.62 | 51.52 | 50.56 | 59.86 | 53.18 |
| GRADE | 53.66 | 57.61 | 52.31 | 50.77 | 53.04 | 54.60 | 50.49 | 62.48 | 57.54 |
| SpecReg | 54.67 | 54.55 | 58.35 | 51.94 | 54.88 | 50.70 | 51.02 | 59.18 | 55.43 |
| A2GNN | 53.84 | 51.17 | 53.45 | 59.31 | 51.15 | 53.43 | 52.64 | 52.42 | 53.24 |
| SimGDA | 58.64 | 57.91 | 59.21 | 58.48 | 55.97 | 57.73 | 52.95 | 60.03 | 53.98 |
| SimGDA+ | 58.64 | 59.97 | 63.01 | 60.34 | 59.44 | 60.06 | 53.67 | 63.61 | 57.69 |

| Models | ES → FR | ES → PT | ES → RU | FR → DE | FR → EN | FR → ES | FR → PT | FR → RU | PT → DE |
|---|---|---|---|---|---|---|---|---|---|
| DANE | 55.36 | 62.22 | 51.71 | 65.58 | 57.48 | 58.45 | 53.49 | 50.43 | 57.66 |
| ACDNE | 52.55 | 53.61 | 51.74 | 54.68 | 53.91 | 52.76 | 54.33 | 51.57 | 52.82 |
| UDAGCN | 52.49 | 62.08 | 52.49 | 56.71 | 56.52 | 59.47 | 58.27 | 54.37 | 53.78 |
| ASN | 53.02 | 55.38 | 51.68 | 55.55 | 53.03 | 53.49 | 51.39 | 51.20 | 55.38 |
| AdaGCN | 53.82 | 62.24 | 52.93 | 59.52 | 57.47 | 61.22 | 56.42 | 51.37 | 57.82 |
| DMGNN | 50.07 | 50.71 | 51.06 | 51.72 | 52.51 | 51.86 | 50.76 | 51.21 | 51.65 |
| CWGCN | 50.92 | 61.59 | 51.83 | 62.45 | 56.55 | 58.15 | 60.44 | 52.87 | 58.30 |
| SAGDA | 53.42 | 52.36 | 50.70 | 54.34 | 54.69 | 51.25 | 52.66 | 50.92 | 55.11 |
| DGDA | 54.95 | 51.72 | 50.95 | 62.74 | 57.39 | 61.17 | 60.90 | 52.12 | 59.19 |
| StruRW | 53.70 | 53.74 | 52.17 | 56.96 | 53.62 | 56.29 | 51.23 | 51.38 | 55.85 |
| KBL | 63.05 | 62.26 | 55.29 | 64.37 | 60.01 | 63.02 | 62.08 | 54.03 | 66.22 |
| JHGDA | 51.62 | 53.98 | 51.01 | 57.57 | 53.98 | 56.94 | 55.36 | 50.92 | 51.35 |
| PairAlign | 54.07 | 50.95 | 52.60 | 53.88 | 53.81 | 53.28 | 51.93 | 52.05 | 54.10 |
| GRADE | 57.72 | 60.46 | 53.56 | 59.25 | 56.58 | 55.08 | 55.71 | 50.34 | 57.54 |
| SpecReg | 55.00 | 50.81 | 51.52 | 59.58 | 55.50 | 54.05 | 50.89 | 51.08 | 56.18 |
| A2GNN | 51.39 | 54.59 | 50.34 | 53.38 | 54.05 | 53.89 | 53.91 | 50.63 | 52.27 |
| SimGDA | 54.05 | 61.91 | 60.35 | 61.09 | 56.00 | 65.66 | 60.93 | 62.65 | 59.47 |
| SimGDA+ | 59.73 | 62.07 | 60.37 | 62.99 | 58.08 | 65.96 | 61.57 | 62.85 | 61.23 |

| Models | PT → EN | PT → ES | PT → FR | PT → RU | RU → DE | RU → EN | RU → ES | RU → FR | RU → PT |
|---|---|---|---|---|---|---|---|---|---|
| DANE | 55.74 | 53.11 | 51.90 | 52.46 | 68.75 | 59.79 | 59.34 | 57.42 | 66.34 |
| ACDNE | 54.34 | 54.46 | 52.51 | 51.52 | 52.46 | 51.32 | 53.05 | 50.96 | 50.78 |
| UDAGCN | 53.01 | 57.08 | 51.63 | 51.19 | 54.74 | 51.32 | 51.30 | 53.21 | 55.47 |
| ASN | 52.13 | 52.03 | 52.80 | 51.71 | 51.97 | 52.87 | 51.86 | 51.24 | 51.77 |
| AdaGCN | 51.80 | 58.28 | 53.98 | 51.23 | 58.35 | 54.78 | 57.76 | 54.35 | 57.23 |
| DMGNN | 52.63 | 51.89 | 50.20 | 50.83 | 52.21 | 52.69 | 52.43 | 50.34 | 51.29 |
| CWGCN | 52.39 | 55.46 | 54.98 | 51.58 | 61.25 | 57.48 | 54.90 | 50.46 | 62.36 |
| SAGDA | 53.92 | 51.58 | 53.92 | 50.37 | 54.84 | 53.62 | 51.41 | 53.83 | 52.67 |
| DGDA | 54.64 | 50.74 | 55.66 | 52.51 | 62.34 | 57.31 | 60.86 | 56.15 | 60.88 |
| StruRW | 52.78 | 53.50 | 53.96 | 50.77 | 52.27 | 51.77 | 52.73 | 50.67 | 51.03 |
| KBL | 56.53 | 64.56 | 52.92 | 53.16 | 59.17 | 55.15 | 59.35 | 58.15 | 59.23 |
| JHGDA | 51.38 | 54.64 | 52.12 | 51.36 | 55.20 | 51.37 | 51.12 | 55.43 | 54.31 |
| PairAlign | 53.97 | 53.79 | 54.42 | 50.88 | 52.29 | 50.94 | 52.87 | 50.90 | 50.80 |
| GRADE | 54.52 | 57.04 | 55.15 | 50.14 | 50.22 | 53.90 | 60.48 | 57.54 | 55.08 |
| SpecReg | 53.12 | 53.34 | 51.71 | 50.13 | 58.77 | 53.59 | 52.17 | 55.42 | 51.01 |
| A2GNN | 50.49 | 53.80 | 50.99 | 51.21 | 51.54 | 52.15 | 54.70 | 51.60 | 53.53 |
| SimGDA | 55.48 | 60.17 | 63.02 | 54.20 | 51.16 | 51.43 | 55.31 | 52.64 | 62.27 |
| SimGDA+ | 57.57 | 61.29 | 63.07 | 55.20 | 61.96 | 56.14 | 59.17 | 56.47 | 62.78 |

Table 12: We evaluated the Macro-F1 score on MAG. OOM indicates out of memory.

| Models | CN → DE | CN → FR | CN → JP | CN → RU | CN → US | DE → CN | DE → FR | DE → JP | DE → RU |
|---|---|---|---|---|---|---|---|---|---|
| DANE | OOM | 12.56 | 19.50 | 11.04 | OOM | OOM | 23.44 | 22.53 | 14.84 |
| ACDNE | 12.18 | 10.41 | 10.08 | 8.57 | 13.40 | 16.08 | 20.99 | 18.07 | 13.95 |
| UDAGCN | 12.84 | 6.85 | 10.69 | 7.32 | 12.23 | 15.01 | 23.26 | 21.82 | 14.48 |
| ASN | 9.52 | OOM | OOM | OOM | 10.64 | 14.67 | 24.08 | 22.60 | 13.99 |
| AdaGCN | 7.63 | 10.51 | 12.36 | 10.65 | 9.30 | 10.75 | 14.79 | 12.85 | 10.47 |
| DMGNN | OOM | 7.64 | 11.18 | 6.99 | OOM | 12.11 | 17.82 | 11.52 | 11.43 |
| CWGCN | OOM | 10.62 | 10.58 | 10.20 | OOM | 11.00 | 13.95 | 12.63 | 9.59 |
| SAGDA | OOM | 6.14 | 9.05 | 6.09 | OOM | OOM | OOM | OOM | OOM |
| DGDA | OOM | OOM | OOM | OOM | OOM | OOM | 19.66 | 18.70 | 11.94 |
| StruRW | 3.32 | 3.54 | 3.75 | 4.59 | 7.69 | 10.55 | 4.94 | 6.08 | 4.63 |
| KBL | 15.79 | 14.03 | 16.53 | 11.69 | 16.98 | 13.14 | 19.51 | 17.26 | 12.60 |
| JHGDA | OOM | OOM | OOM | OOM | OOM | OOM | 25.10 | 21.69 | 12.52 |
| PairAlign | 10.52 | 20.90 | 18.70 | 12.13 | 9.33 | 11.53 | 17.61 | 13.80 | 12.45 |
| GRADE | 11.72 | 11.06 | 13.18 | 10.14 | 12.54 | 11.21 | 17.30 | 14.06 | 10.43 |
| SpecReg | 19.42 | 12.11 | 13.85 | 11.64 | 22.23 | 23.22 | 30.80 | 28.13 | 17.89 |
| A2GNN | 22.59 | 18.44 | 22.22 | 13.37 | 25.23 | 19.75 | 24.15 | 24.16 | 12.19 |
| SimGDA | 12.20 | 11.42 | 13.83 | 11.40 | 14.09 | 14.55 | 19.53 | 17.53 | 13.28 |
| SimGDA+ | 21.28 | 18.25 | 22.50 | 13.96 | 23.16 | 18.50 | 25.39 | 24.60 | 15.47 |

| Models | DE → US | FR → CN | FR → DE | FR → RU | FR → US | JP → CN | JP → DE | JP → US | RU → CN |
|---|---|---|---|---|---|---|---|---|---|
| DANE | OOM | 15.77 | 22.11 | 12.92 | 16.38 | 16.59 | 20.21 | OOM | 7.35 |
| ACDNE | 18.04 | 14.20 | 20.47 | 14.60 | 14.79 | 14.89 | 15.79 | 15.96 | 6.73 |
| UDAGCN | 25.24 | 16.27 | 25.97 | 8.47 | 24.25 | 17.34 | 21.31 | 18.57 | 8.70 |
| ASN | 23.11 | 15.22 | 25.62 | 10.92 | 21.22 | 15.36 | 22.27 | 20.77 | 8.88 |
| AdaGCN | 10.48 | 9.27 | 12.73 | 12.34 | 9.17 | 9.99 | 9.97 | 9.63 | 4.36 |
| DMGNN | 12.65 | 11.52 | 16.84 | 9.89 | 12.94 | 11.62 | 14.45 | 11.39 | 4.10 |
| CWGCN | 10.78 | OOM | 0.19 | 11.16 | 10.33 | 9.93 | 9.39 | 9.44 | 4.74 |
| SAGDA | OOM | OOM | OOM | 2.98 | OOM | OOM | OOM | OOM | OOM |
| DGDA | OOM | OOM | 15.80 | OOM | OOM | OOM | 12.51 | OOM | OOM |
| StruRW | 12.16 | 15.67 | 15.27 | 14.65 | 25.45 | 8.30 | 6.59 | 20.05 | 5.65 |
| KBL | 13.94 | 14.08 | 18.06 | 13.88 | 13.39 | 14.61 | 16.24 | 17.40 | 12.49 |
| JHGDA | OOM | OOM | 24.23 | 12.88 | OOM | OOM | 21.96 | 0.00 | 0.00 |
| PairAlign | 13.44 | 8.67 | 15.78 | 12.49 | 11.65 | 12.99 | 12.49 | 12.87 | 4.51 |
| GRADE | 12.52 | 10.92 | 16.94 | 9.56 | 12.57 | 11.98 | 11.64 | 13.95 | 4.09 |
| SpecReg | 29.09 | 22.49 | 31.57 | 14.33 | 28.01 | 25.54 | 28.97 | 30.30 | 17.65 |
| A2GNN | 27.67 | 20.95 | 28.57 | 16.13 | 28.51 | 21.71 | 25.53 | 27.14 | 18.91 |
| SimGDA | 16.84 | 13.53 | 18.83 | 12.51 | 16.01 | 14.21 | 14.66 | 15.04 | 6.36 |
| SimGDA+ | 26.37 | 17.72 | 28.91 | 15.09 | 26.35 | 19.99 | 25.43 | 24.22 | 15.05 |

| Models | RU → DE | RU → FR | RU → JP | RU → US | US → CN | US → DE | US → FR | US → JP | US → RU |
|---|---|---|---|---|---|---|---|---|---|
| DANE | 7.75 | 6.17 | 8.47 | 6.57 | OOM | OOM | 22.27 | OOM | OOM |
| ACDNE | 5.98 | 6.12 | 7.42 | 5.93 | 20.63 | 22.48 | 19.65 | 23.90 | 14.86 |
| UDAGCN | 8.42 | 7.96 | 8.92 | 8.38 | 19.01 | 28.24 | 25.17 | 25.80 | 15.37 |
| ASN | 9.69 | 9.73 | 8.99 | 10.88 | 15.99 | 24.52 | 21.09 | 22.93 | 12.27 |
| AdaGCN | 3.59 | 3.73 | 4.25 | 3.26 | 14.80 | 15.95 | 14.40 | 18.34 | 10.99 |
| DMGNN | 3.69 | 4.34 | 4.42 | 3.22 | OOM | OOM | OOM | OOM | OOM |
| CWGCN | 3.32 | 3.76 | 4.60 | 4.61 | 15.11 | 15.54 | 13.41 | 16.85 | 11.59 |
| SAGDA | OOM | 5.61 | OOM | OOM | 0.00 | 0.00 | 0.00 | 0.00 | 0.00 |
| DGDA | 4.17 | 4.52 | 6.98 | OOM | 0.00 | 0.00 | 0.00 | 0.00 | 0.00 |
| StruRW | 4.88 | 3.44 | 4.31 | 4.79 | 8.44 | 10.61 | 3.44 | 6.37 | 7.44 |
| KBL | 11.52 | 9.35 | 12.92 | 11.49 | 16.62 | 19.67 | 17.90 | 19.25 | 12.69 |
| JHGDA | 19.85 | 16.67 | 19.15 | OOM | OOM | OOM | OOM | OOM | OOM |
| PairAlign | 3.61 | 4.21 | 4.60 | 3.51 | 16.12 | 17.85 | 16.66 | 19.28 | 11.93 |
| GRADE | 3.32 | 3.47 | 4.01 | 3.18 | 16.30 | 16.28 | 17.02 | 21.42 | 11.83 |
| SpecReg | 18.66 | 16.51 | 21.35 | 16.02 | 26.23 | 31.82 | 28.81 | 30.12 | 16.23 |
| A2GNN | 22.67 | 20.36 | 20.90 | 22.24 | 21.47 | 27.42 | 25.29 | 25.43 | 13.06 |
| SimGDA | 6.09 | 6.21 | 7.48 | 6.33 | 18.37 | 11.31 | 10.59 | 22.19 | 14.31 |
| SimGDA+ | 20.35 | 16.84 | 20.06 | 20.65 | 21.17 | 26.94 | 24.23 | 25.58 | 15.62 |

Table 13: Domain shifts statistics of each task.

| Dataset | Source | Target | Feature Shift | Structure Shift | Label Shift |
|---|---|---|---|---|---|
| Blog | Blog1 | Blog2 | 0.0140 | 0.0802 | 0.253 |
| | Blog2 | Blog1 | 0.0137 | 0.0802 | 0.258 |
| Airport | USA | BRAZIL | 0.0514 | 0.2331 | 0.065 |
| | USA | EUROPE | 0.0913 | 0.3983 | 0.005 |
| | BRAZIL | USA | 0.0523 | 0.2331 | 0.066 |
| | BRAZIL | EUROPE | 0.0549 | 0.1993 | 0.035 |
| | EUROPE | USA | 0.1000 | 0.3983 | 0.005 |
| | EUROPE | BRAZIL | 0.0582 | 0.1993 | 0.034 |
| ArnetMiner | DBLPv7 | ACMv9 | 0.0312 | 0.2327 | 0.997 |
| | DBLPv7 | Citationv1 | 0.0245 | 0.1965 | 1.643 |
| | ACMv9 | DBLPv7 | 0.0305 | 0.2327 | 1.062 |
| | ACMv9 | Citationv1 | 0.0163 | 0.1931 | 0.780 |
| | Citationv1 | DBLPv7 | 0.0244 | 0.1965 | 1.624 |
| | Citationv1 | ACMv9 | 0.0166 | 0.1931 | 0.805 |
| Twitch | EN | DE | 0.0493 | 0.1486 | 0.715 |
| | EN | FR | 0.0440 | 0.3148 | 6.449 |
| | EN | RU | 0.0368 | 0.5960 | 20.578 |
| | EN | ES | 0.0530 | 0.3836 | 13.883 |
| | EN | PT | 0.0790 | 0.3374 | 8.330 |
| | DE | EN | 0.0478 | 0.1486 | 0.707 |
| | DE | FR | 0.0408 | 0.4635 | 11.403 |
| | DE | RU | 0.0387 | 0.7446 | 28.985 |
| | DE | ES | 0.0283 | 0.5323 | 20.866 |
| | DE | PT | 0.0391 | 0.4860 | 13.871 |
| | FR | EN | 0.0463 | 0.3148 | 6.315 |
| | FR | DE | 0.0383 | 0.4635 | 11.302 |
| | FR | RU | 0.0503 | 0.2811 | 3.754 |
| | FR | ES | 0.0432 | 0.0688 | 1.335 |
| | FR | PT | 0.0733 | 0.0226 | 0.115 |
| | RU | EN | 0.0369 | 0.5960 | 18.693 |
| | RU | DE | 0.0355 | 0.7446 | 26.658 |
| | RU | FR | 0.0542 | 0.2811 | 3.479 |
| | RU | ES | 0.0426 | 0.2124 | 0.562 |
| | RU | PT | 0.0551 | 0.2586 | 2.363 |
| | ES | EN | 0.0525 | 0.3836 | 13.080 |
| | ES | DE | 0.0282 | 0.5323 | 19.901 |
| | ES | FR | 0.0406 | 0.0688 | 1.284 |
| | ES | RU | 0.0460 | 0.2124 | 0.583 |
| | ES | PT | 0.0320 | 0.0462 | 0.640 |
| | PT | EN | 0.0776 | 0.3374 | 8.080 |
| | PT | DE | 0.0407 | 0.4860 | 13.620 |
| | PT | FR | 0.0713 | 0.0226 | 0.114 |
| | PT | RU | 0.0554 | 0.2586 | 2.526 |
| | PT | ES | 0.0311 | 0.0462 | 0.660 |
| MAG | CN | DE | 0.0750 | 0.3608 | 33.807 |
| | CN | FR | 0.0773 | 0.3902 | 26.427 |
| | CN | JP | 0.0451 | 0.2775 | 16.382 |
| | CN | RU | 0.0779 | 0.5454 | 30.058 |
| | CN | US | 0.0781 | 0.2858 | 28.992 |
| | DE | CN | 0.0727 | 0.3608 | 46.271 |
| | DE | FR | 0.0213 | 0.2041 | 2.316 |
| | DE | JP | 0.0464 | 0.3278 | 22.811 |
| | DE | RU | 0.0419 | 0.4778 | 48.632 |
| | DE | US | 0.0179 | 0.3561 | 16.266 |
| | FR | CN | 0.0702 | 0.3902 | 46.780 |
| | FR | DE | 0.0196 | 0.2041 | 2.241 |
| | FR | JP | 0.0508 | 0.3815 | 30.343 |
| | FR | RU | 0.0382 | 0.5091 | 49.558 |
| | FR | US | 0.0187 | 0.4210 | 23.644 |
| | JP | CN | 0.0486 | 0.2775 | 14.352 |
| | JP | DE | 0.0391 | 0.3278 | 12.240 |
| | JP | FR | 0.0467 | 0.3815 | 13.597 |
| | JP | RU | 0.0513 | 0.4968 | 27.544 |
| | JP | US | 0.0540 | 0.2893 | 8.235 |
| | RU | CN | 0.0776 | 0.5454 | 18.701 |
| | RU | DE | 0.0442 | 0.4778 | 31.345 |
| | RU | FR | 0.0416 | 0.5091 | 28.260 |
| | RU | JP | 0.0524 | 0.4968 | 18.269 |
| | RU | US | 0.0517 | 0.6171 | 35.979 |
| | US | CN | 0.0832 | 0.2858 | 35.273 |
| | US | DE | 0.0206 | 0.3561 | 14.702 |
| | US | FR | 0.0197 | 0.4210 | 19.245 |
| | US | JP | 0.0431 | 0.2893 | 10.104 |
| | US | RU | 0.0567 | 0.6171 | 60.803 |

Table 14: Parameter search space list.

| Dataset | Models | | Hyperparameter | Search Space |
|---|---|---|---|---|
| Airport | SimGDA+ | SimGDA | learning rate | [0.0001, 0.0005, 0.001, 0.005] |
| | | | weight decay | [0.0001, 0.0005, 0.001, 0.005] |
| | | | momentum | [0.01, 0.99] |
| | | | backbone | gcn |
| | | | backbone layers | [1, 3, 4, 5] |
| | | | dropout ratio | 0.5 |
| | | | feature dimension | 128 |
| | | | alpha | [0.5, 1] |
| | | | epochs | 200 |
| | | SimGDA + IM | beta | [0, 0.05, 0.1, 0.5] |
| | | | epochs | 200 |
| | | SimGDA + AE | beta | [0, 0.05, 0.1, 0.5] |
| | | | decoder dropout | 0.1 |
| | | SimGDA + CL | beta | [0, 0.05, 0.1, 0.5] |
| | | | epochs | 500 |
| | | | augment dropout | [0.1, 0.9] |
| | | | temperature | [0.1, 0.9] |
| | GDABench Baselines | | learning rate | [0.0001, 0.001, 0.003] |
| | | | weight decay | [0.0001, 0.001, 0.003, 0.01] |
| | | | backbone layers | [1, 2, 3, 4, 5] |
| | | | dropout ratio | [0.1, 0.2, 0.3, 0.4, 0.5] |
| | | | feature dimension | 128 |
| | | | epochs | [100, 200, 400] |
| Blog | SimGDA+ | SimGDA | learning rate | [0.0001, 0.0005] |
| | | | weight decay | [0.001, 0.005] |
| | | | momentum | [0.01, 0.99] |
| | | | backbone | gcn |
| | | | backbone layers | [1, 2, 3, 4, 5] |
| | | | dropout ratio | 0.5 |
| | | | feature dimension | 128 |
| | | | alpha | [0.5, 1] |
| | | | epochs | 200 |
| | | SimGDA + AE | beta | [0, 0.05] |
| | | | decoder dropout | 0.1 |
| | GDABench Baselines | | learning rate | [0.0001,0.0003, 0.001] |
| | | | weight decay | [0.001, 0.003, 0.01] |
| | | | backbone layers | [1, 2, 3, 4] |
| | | | dropout ratio | [0.1, 0.2, 0.3, 0.4, 0.5] |
| | | | feature dimension | 128 |
| | | | epochs | [200, 300, 400] |
| ArnetMiner | SimGDA+ | SimGDA | learning rate | [0.0001, 0.0005, 0.001, 0.005] |
| | | | weight decay | [0.0005, 0.001, 0.005] |
| | | | momentum | [0.01, 0.99] |
| | | | backbone | gcn |
| | | | backbone layers | [1, 2, 3, 4, 5] |
| | | | dropout ratio | 0.5 |
| | | | feature dimension | 128 |
| | | | alpha | [0.5, 1] |
| | | | epochs | 200 |
| | | SimGDA + IM | beta | [0.5, 1] |
| | | | epochs | 200 |
| | GDABench Baselines | | learning rate | [0.0001, 0.001, 0.003, 0.01] |
| | | | weight decay | [0.0001, 0.001, 0.003, 0.01] |
| | | | backbone layers | [1, 2, 3, 4, 5] |
| | | | dropout ratio | [0.1, 0.2, 0.3, 0.4, 0.5] |
| | | | feature dimension | 128 |
| | | | epochs | [100, 200, 400, 800] |
| Twitch | SimGDA+ | SimGDA | learning rate | [0.0001, 0.0005, 0.001, 0.005] |
| | | | weight decay | [0.0001, 0.0005, 0.001, 0.005] |
| | | | momentum | [0.01, 0.99] |
| | | | backbone | gcn |
| | | | backbone layers | [1, 2, 3, 4, 5] |
| | | | dropout ratio | 0.5 |
| | | | feature dimension | 128 |
| | | | alpha | [0.5, 1] |
| | | | epochs | 200 |
| | | SimGDA + AE | beta | [0, 0.05, 0.1, 0.2, 0.5] |
| | | | decoder dropout | [0.1, 0.9] |
| | | SimGDA + CL | beta | [0, 0.05, 0.1, 0.5, 1, 1.5] |
| | | | epochs | 500 |
| | | | augment dropout | [0.1, 0.9] |
| | | | temperature | [0.1, 0.9] |
| | GDABench Baselines | | learning rate | [0.0001, 0.001, 0.003] |
| | | | weight decay | [0.0001, 0.001, 0.003, 0.01] |
| | | | backbone layers | [1, 2, 3, 4, 5] |
| | | | dropout ratio | [0.1, 0.2, 0.3, 0.4, 0.5] |
| | | | feature dimension | 128 |
| | | | epochs | [100, 200, 400] |
| MAG | SimGDA+ | SimGDA | learning rate | [0.0001, 0.0005, 0.001, 0.005] |
| | | | weight decay | [0.0001, 0.0005, 0.001, 0.005] |
| | | | momentum | [0.01, 0.99] |
| | | | backbone | gcn |
| | | | backbone layers | [1, 2, 3, 4, 5] |
| | | | dropout ratio | 0.5 |
| | | | feature dimension | 128 |
| | | | alpha | [0.5, 1] |
| | | | epochs | 200 |
| | GDABench Baselines | | learning rate | [0.0001, 0.001, 0.003] |
| | | | weight decay | [0.0001, 0.001, 0.003] |
| | | | backbone layers | [1, 2, 3] |
| | | | dropout ratio | [0.1, 0.2, 0.3, 0.4, 0.5] |
| | | | feature dimension | 300 |
| | | | epochs | [200, 400, 600, 800] |

Table 15: To evaluate the baselines on graph-level shifts, we compared the Micro-F1 scores of each model on the Proteins, Mutagenicity, and Frankenstein datasets. The best results are highlighted in **bold**, and the second-best results are underlined.

| Models | Proteins | | Mutagenicity | | Frankenstein | |
|---|---|---|---|---|---|---|
| | P1 → P2 | P2 → P1 | M1 → M2 | M2 → M1 | F1 → F2 | F2 → F1 |
| DANE | **60.14** $_{\pm 3.58}$ | 75.66 $_{\pm 0.98}$ | 67.25 $_{\pm 0.14}$ | **76.92** $_{\pm 0.35}$ | 54.77 $_{\pm 0.53}$ | 56.96 $_{\pm 2.89}$ |
| UDAGCN | 53.50 $_{\pm 2.42}$ | 73.14 $_{\pm 4.29}$ | 58.11 $_{\pm 0.58}$ | 65.34 $_{\pm 0.55}$ | 52.48 $_{\pm 0.32}$ | 52.37 $_{\pm 1.38}$ |
| AdaGCN | 52.60 $_{\pm 0.78}$ | **78.12** $_{\pm 0.37}$ | 58.89 $_{\pm 0.06}$ | 56.18 $_{\pm 0.02}$ | 56.28 $_{\pm 0.75}$ | 53.01 $_{\pm 3.63}$ |
| CWGCN | 50.45 $_{\pm 4.81}$ | 44.84 $_{\pm 8.20}$ | 55.60 $_{\pm 1.27}$ | 56.72 $_{\pm 0.67}$ | 49.76 $_{\pm 0.27}$ | 51.92 $_{\pm 0.71}$ |
| SAGDA | 53.14 $_{\pm 4.80}$ | 46.22 $_{\pm 2.99}$ | 57.06 $_{\pm 3.54}$ | 56.00 $_{\pm 8.85}$ | 50.35 $_{\pm 0.26}$ | 51.01 $_{\pm 8.37}$ |
| GRADE | 43.93 $_{\pm 0.31}$ | 76.80 $_{\pm 0.29}$ | **69.00** $_{\pm 0.22}$ | 76.57 $_{\pm 0.31}$ | **57.54** $_{\pm 1.09}$ | **58.39** $_{\pm 4.57}$ |
| A2GNN | 51.70 $_{\pm 1.54}$ | 69.65 $_{\pm 4.21}$ | 56.83 $_{\pm 0.19}$ | 58.88 $_{\pm 1.23}$ | 50.43 $_{\pm 0.69}$ | 48.99 $_{\pm 3.97}$ |

Table 16: To evaluate the baselines on graph-level shifts, we compared the Macro-F1 scores of each model on the Proteins, Mutagenicity, and Frankenstein datasets. The best results are highlighted in **bold**, and the second-best results are underlined.

| Models | Proteins | | Mutagenicity | | Frankenstein | |
|---|---|---|---|---|---|---|
| | P1 → P2 | P2 → P1 | M1 → M2 | M2 → M1 | F1 → F2 | F2 → F1 |
| DANE | **59.14** $_{\pm 3.06}$ | 56.30 $_{\pm 6.09}$ | 67.11 $_{\pm 0.17}$ | **76.50** $_{\pm 0.35}$ | 52.24 $_{\pm 1.02}$ | 54.94 $_{\pm 2.13}$ |
| UDAGCN | 53.15 $_{\pm 2.74}$ | 50.19 $_{\pm 1.20}$ | 56.71 $_{\pm 0.61}$ | 63.35 $_{\pm 0.56}$ | 50.06 $_{\pm 0.64}$ | 52.32 $_{\pm 1.40}$ |
| AdaGCN | 49.33 $_{\pm 1.62}$ | 57.99 $_{\pm 2.82}$ | 58.00 $_{\pm 0.10}$ | 35.97 $_{\pm 0.10}$ | 55.99 $_{\pm 0.94}$ | 51.76 $_{\pm 4.43}$ |
| CWGCN | 40.57 $_{\pm 3.13}$ | 42.75 $_{\pm 6.01}$ | 39.00 $_{\pm 4.96}$ | 37.32 $_{\pm 1.66}$ | 39.46 $_{\pm 0.22}$ | 51.68 $_{\pm 0.67}$ |
| SAGDA | 46.65 $_{\pm 6.14}$ | 33.42 $_{\pm 1.01}$ | 56.26 $_{\pm 3.74}$ | 54.95 $_{\pm 8.22}$ | 36.89 $_{\pm 4.93}$ | 38.03 $_{\pm 6.81}$ |
| GRADE | 32.23 $_{\pm 0.86}$ | 50.52 $_{\pm 1.77}$ | **68.98** $_{\pm 0.21}$ | 76.32 $_{\pm 0.26}$ | **56.93** $_{\pm 1.80}$ | **54.98** $_{\pm 2.62}$ |
| A2GNN | 47.71 $_{\pm 3.22}$ | **58.85** $_{\pm 1.16}$ | 55.42 $_{\pm 0.10}$ | 50.17 $_{\pm 1.59}$ | 46.97 $_{\pm 0.96}$ | 43.33 $_{\pm 1.87}$ |

vice versa. Consequently, this variability complicates the task of assessing how well these models will perform when deployed in diverse and complex real-world scenarios where both node-level and graph-level information may be critical.

***Domain adaptive message passing methods demonstrate superior and consistency performance across a wide range of datasets and tasks.*** As shown in Table 3, 9, 16 and 15, methods designed based on the inherent properties of GNN achieves the top-three best performance in 8 tasks out of 12 node-level tasks and top-two best performance in 5 tasks out of 6 graph-level tasks. This observation verified our findings that establishing domain adaptation principles by leveraging inherent properties of GNN can result in an effective and efficient approach to addressing the challenges of domain variability in graph datasets.

To summarize, our observations underscore the importance of leveraging the intrinsic properties of GNNs to devise effective domain adaptation strategies, which not only enhances performance but also ensures consistency in real-world applications.

# F   Discussion

## F.1   How these findings generalize to real-world scenarios

Our benchmark includes a range of datasets with varying characteristics to capture different aspects of graph domain adaptation. This diversity aims to provide a broad perspective on the applicability of our methods. In real-world scenarios, applying graph adaptation methods effectively involves several key considerations: Firstly, it is imperative to develop tailored strategies specifically designed to address the structural shifts observed in graphs. For example, if a graph is dynamic and changing overtime, it is crucial to accord greater attention to its evolving structure. Secondly, recognizing the importance of the aggregation scope and aggregation architecture in GNNs' transferability within unsupervised graph domain adaptation (UGDA) are crucial. In real-world graphs, noise is inevitable, hence, strategically selecting effective neighbors not only improve performance but also avoid noise.

Thirdly, by leveraging the properties of GNNs that make them inherently adaptable to changes in graph structure and data distribution, we can develop simple yet highly effective models.

### F.2  A broader discussion on DA problem and other related UGDA scenarios

**UDA vs UGDA.**  Unsupervised domain adaptation (UDA) entails transferring knowledge from a labeled source domain to an unlabeled target domain. A prevalent strategy in domain adaptation is to reduce domain discrepancies while learning domain-invariant representations, a method that has seen considerable success in the fields of computer vision and natural language processing. However, these techniques typically operate under the assumption that inputs are independently and identically distributed (IID), making them unsuitable for tasks involving non-IID data, such as node classification in graph-structured datasets.

**UGDA vs muti-domain UGDA.**  Muti-domain UGDA extends the concept of domain adaptation to situations where there are multiple source domains and a single target domain. This approach aims to learn a model that can generalize well across multiple source domains, and then adapt it to perform well on the target domain. Compared to standard UGDA, multi-domain UGDA can enhance generalization by leveraging the diversity of multiple source domains. However, it may require more complex models and additional computational resources.

**UGDA vs source-free UGDA.**  Source-free UGDA advances domain adaptation by tackling the challenge of adapting models without access to labeled data from the source domains. This setting is more challenging as it involves learning to transfer knowledge without explicit supervision. Source-free UGDA methods often employ techniques such as self-training or consistency regularization to adapt the model to the target domain. Compared to UGDA, source-free UGDA may be more sensitive to domain shift and require careful selection of adaptation techniques.

