# OpenReview forum: "Revisiting, Benchmarking and Understanding Unsupervised Graph Domain Adaptation"
_NeurIPS.cc/2024/Datasets_and_Benchmarks_Track — NeurIPS 2024 Track Datasets and Benchmarks Poster_

### Official Review · Reviewer_TozF · 2024-06-14

**Rating:** 8
**Confidence:** 5
**Correctness:** Yes
**Clarity:** Yes

**Review:**

Strength
1. GDABench includes a wide range of state-of-the-art UGDA models and diverse datasets, providing a thorough evaluation of model performance across multiple scenarios, which is essential for understanding the strengths and weaknesses of different approaches.

2. The benchmark addresses real-world challenges such as significant distribution shifts in node attributes, graph structures, and label proportions. This focus ensures that the benchmark is relevant and applicable to practical problems in UGDA.

3. GDABench offers standardized evaluation procedures and a publicly available library, PyGDA, which promotes transparency, reproducibility, and ease of use for researchers, encouraging further research and development in the field.


Weakness & discussion

1. The paper is overall good, which points out that the GNN design can help to solve the structural shift. [1] also illustrates the phenomenon both theoretically and empirically. [1] shows that the reason why deeper GNN can perform well is the mitigation of the structural disparity. One additional thing for improvement is to take the heterophily graphs into the discussion, instead of the majority focus on the homophily graphs

2. As mentioned in the paper, the feature shift is one important source of the domain gap. Notably, most of the introduced dataset is text-attributed graphs while utilize Bow or word2vec as the feature encoder. It is questionable whether the distribution gap exists after utilizing the LLM as the feature encoder.

3. The paper discusses the wide range of distribution shifts, however, how to evaluate them is not shown in the main contents. I think this part could be important for this community

4. One core limitation of the domain adaptation is the requirement for the same label space. I am very interested in the discussion about the practice value of the UGDA. One potential I think about is similar to the ogbn-arxiv data split, utilizing time as different domains.

5. The paper does a good job of categorizing the basic UGDA scenario. It could be better to see a more comprehensive discussion on other related UGDA scenarios[3], multi-domain UGDA[5], and source-free UGDA[2,4].

[1] Mao, Haitao, et al. "Demystifying Structural Disparity in Graph Neural Networks: Can One Size Fit All?." Advances in Neural Information Processing Systems 36 (2024).
[2] Zhang, Zhen, et al. "Collaborate to Adapt: Source-Free Graph Domain Adaptation via Bi-directional Adaptation." Proceedings of the ACM on Web Conference 2024. 2024.
[3] Shi, Boshen, et al. "Graph Domain Adaptation: Challenges, Progress and Prospects." arXiv preprint arXiv:2402.00904 (2024).
[4] Mao, Haitao, et al. "Source Free Graph Unsupervised Domain Adaptation." Proceedings of the 17th ACM International Conference on Web Search and Data Mining. 2024.
[5] He, Hui, et al. "MSDS: A Novel Framework for Multi-Source Data Selection Based Cross-Network Node Classification." IEEE Transactions on Knowledge and Data Engineering (2023).

**Strengths:**

1. GDABench includes a wide range of state-of-the-art UGDA models and diverse datasets, providing a thorough evaluation of model performance across multiple scenarios, which is essential for understanding the strengths and weaknesses of different approaches.

2. The benchmark addresses real-world challenges such as significant distribution shifts in node attributes, graph structures, and label proportions. This focus ensures that the benchmark is relevant and applicable to practical problems in UGDA.

3. GDABench offers standardized evaluation procedures and a publicly available library, PyGDA, which promotes transparency, reproducibility, and ease of use for researchers, encouraging further research and development in the field.

**Additional Feedback:**

No

**Documentation:**

Yes

**Opportunities For Improvement:**

1. The paper is overall good, which points out that the GNN design can help to solve the structural shift. [1] also illustrates the phenomenon both theoretically and empirically. [1] shows that the reason why deeper GNN can perform well is the mitigation of the structural disparity. One additional thing for improvement is to take the heterophily graphs into the discussion, instead of the majority focus on the homophily graphs

2. As mentioned in the paper, the feature shift is one important source of the domain gap. Notably, most of the introduced dataset is text-attributed graphs while utilize Bow or word2vec as the feature encoder. It is questionable whether the distribution gap exists after utilizing the LLM as the feature encoder.

3. The paper discusses the wide range of distribution shifts, however, how to evaluate them is not shown in the main contents. I think this part could be important for this community

4. One core limitation of the domain adaptation is the requirement for the same label space. I am very interested in the discussion about the practice value of the UGDA. One potential I think about is similar to the ogbn-arxiv data split, utilizing time as different domains.

5. The paper does a good job of categorizing the basic UGDA scenario. It could be better to see a more comprehensive discussion on other related UGDA scenarios[3], multi-domain UGDA[5], and source-free UGDA[2,4].

[1] Mao, Haitao, et al. "Demystifying Structural Disparity in Graph Neural Networks: Can One Size Fit All?." Advances in Neural Information Processing Systems 36 (2024).
[2] Zhang, Zhen, et al. "Collaborate to Adapt: Source-Free Graph Domain Adaptation via Bi-directional Adaptation." Proceedings of the ACM on Web Conference 2024. 2024.
[3] Shi, Boshen, et al. "Graph Domain Adaptation: Challenges, Progress and Prospects." arXiv preprint arXiv:2402.00904 (2024).
[4] Mao, Haitao, et al. "Source Free Graph Unsupervised Domain Adaptation." Proceedings of the 17th ACM International Conference on Web Search and Data Mining. 2024.
[5] He, Hui, et al. "MSDS: A Novel Framework for Multi-Source Data Selection Based Cross-Network Node Classification." IEEE Transactions on Knowledge and Data Engineering (2023).

**Relation To Prior Work:**

See in weakness

**Summary And Contributions:**

This paper introduces GDABench, a comprehensive benchmark for Unsupervised Graph Domain Adaptation (UGDA). UGDA involves transferring knowledge from a labeled source graph to an unlabeled target graph despite domain discrepancies. GDABench includes 16 state-of-the-art UGDA models and evaluates them across five diverse datasets, resulting in 74 adaptation tasks. The critical role of aggregation mechanisms in GNNs for enhancing transferability.

---

> ### Author Rebuttal · Authors · 2024-08-17
>
> Dear Reviewer TozF,
>
> Thank you for your insightful comments and suggestions! We appreciate your encouragement to broaden our research，which would  improve the overall quality of our work. Below, we provide our detailed response to your feedback.
>
> ---
>
> **Q1.** *The paper is overall good, which points out that the GNN design can help to solve the structural shift. [1] also illustrates the phenomenon both theoretically and empirically. [1] shows that the reason why deeper GNN can perform well is the mitigation of the structural disparity. One additional thing for improvement is to take the heterophily graphs into the discussion, instead of the majority focus on the homophily graphs.*
>
> **A1.** We appreciate you pointing out [1] and we will cite it in the revised version. Regarding your suggestion to consider heterophily graphs, we would like to clarify that we have indeed taken the difference in homophily between source and target graphs into account. For example, in task U → E (Airport dataset), the source graph U is more homogeneous (0.6978), while the target graph E is more heterophilous (0.4048). We present detailed results of homophily levels in Table 4 of the Appendix. The degree of homophily difference is integral to our concept of structure shift, and further details on its calculation can be found in Appendix B.2 or in reference [6].
>
> ---
> **Q2.** *As mentioned in the paper, the feature shift is one important source of the domain gap. Notably, most of the introduced dataset is text-attributed graphs while utilize Bow or word2vec as the feature encoder. It is questionable whether the distribution gap exists after utilizing the LLM as the feature encoder.*
>
> **A2.** Thank you for raising this important point. We agree that utilizing large language models (LLMs) could be a promising strategy to address feature shift in text-attributed graphs. LLMs have the potential to preprocess text features in an unified way that narrows the feature shift, allowing us to focus more on structure and label shifts. However, it's important to note that this approach falls outside the scope of our current work, which primarily focuses on domain adaptation without the use of LLMs. We would be very interested in exploring this in the future.
>
> ---
> **Q3.** *The paper discusses the wide range of distribution shifts, however, how to evaluate them is not shown in the main contents. I think this part could be important for this community.*
>
> **A3.** Thank you for highlighting this. We use MMD, CSS [6], Kullback-Leibler Divergence to characterize the degree of feature shift, structure shift and label shift, which is detailed in Appendix B.2 and supported by numerical values in Tables 4, 5, 9, and Figure 4. We will move more contents from Appendix to the main text in the revised version.
>
> ---
> **Q4.** *One core limitation of the domain adaptation is the requirement for the same label space. I am very interested in the discussion about the practice value of the UGDA. One potential I think about is similar to the ogbn-arxiv data split, utilizing time as different domains.*
>
> **A4.** Thanks for raising this point. While the requirement for the same label space could be a limitation,  UGDA can be particularly useful in various scenarios. Notably, it can be employed to mitigate cold-start problems and reduce the need for extensive label annotations. By leveraging knowledge from source domains, UGDA effectively addresses the challenge of insufficient labeled data in new or sparse target domains, thereby alleviating issues related to cold-start scenarios. Besides the example you mentioned, here are more applications and examples:
> * Social Network Analysis: Adapting user behavior models from one social network (e.g., Facebook) to another (e.g., Twitter). Users might have similar patterns of interactions, but the way these platforms represent connections and interactions can differ.
> * Transportation Analysis: Adapting traffic flow models from one city to another to predict congestion and optimize traffic signals, even if the road network structures differ.
> * Financial Analysis: Adapting financial transaction patterns observed in one region or market to improve fraud detection models in another market.
>
> Overall, graph domain adaptation can facilitate better performance in diverse applications by allowing for the transfer of knowledge and insights across different yet related graph structures. Our datasets also encompass various downstream applications. We will add the discussion about the practice value of the UGDA in our revised version.

---

> > ### Comment · Reviewer_TozF · 2024-08-17
> > **Response**
> >
> > Thanks for your detailed response.
> >
> > For Q1, I think one additional point is that you can add a few sentences to describe how UDA algorithms performs under the homophily ratio shift, for a more comprehensive discussion.
> >
> > For Q2, I still recommend some experiments to take LLM embedding into consideration or provide the original text data for further investigation. I am not to say the authors must conduct experiments during the short rebuttal period, but I would like to see more in the next paper version. I strongly believe such revision leads to a more rigorous and convincing analysis and totally shift the graph UDA into the new era.
> >
> > For Q4, I think the transportation and financial ones are good examples. For social network, I dis-agree with this one as different domains have different label sets in most cases.
> >
> > Overall, authors have solved most of my concerns, I will not raise my score due to the Q2. I will raise my confidence to 5 and champion for the paper.
> >
> > One additional thing is that I also read other reviewer's comment.  I find reviewer vbxq is high likely to be a chatgpt-generated response. I will report this to AC to chase for a fair review process.

---

> > > ### Author Response · Authors · 2024-08-24
> > > **Response of Q1**
> > >
> > > Thank you for your prompt feedback and for raising your confidence to 5! We appreciate your kind support. Below is our detailed response to your comments.
> > >
> > > ---
> > >
> > > **Q1.** *I think one additional point is that you can add a few sentences to describe how UDA algorithms performs under the homophily ratio shift, for a more comprehensive discussion.*
> > >
> > > **A1.** Thanks for this suggestion. In general, our findings indicate that as the homophily ratio shift increases, the performance gains tend to diminish. This is primarily because larger shifts in homophily ratios introduce greater challenges for the model, making it more difficult to effectively adapt across domains. The greater the disparity, the more complex the task becomes, which can hinder the model's ability to achieve significant performance improvements. For instance, in ArnetMiner dataset task A (homo ratio: 0.79) -> D (homo ratio: 0.81) and task A (homo ratio: 0.79) -> C (homo ratio: 0.86), we observed that the performance gains of A->D are generally larger than A->C when compared to GCN without adaptation as a baseline. This is also observed in the inverse scenarios. We will include these discussions in the final version.

---

> > > > ### Author Response · Authors · 2024-08-24
> > > > **Response of Q2 (Part 1)**
> > > >
> > > > **Q2.** *I still recommend some experiments to take LLM embedding into consideration or provide the original text data for further investigation. I am not to say the authors must conduct experiments during the short rebuttal period, but I would like to see more in the next paper version. I strongly believe such revision leads to a more rigorous and convincing analysis and totally shift the graph UDA into the new era.*
> > > >
> > > > **A2.**  Thank you for your suggestion for considering LLM embedding in experiments. We agree and are happy to report that we were able to execute all the suggested experiments.
> > > >
> > > > To investigate whether the distribution gap narrows after utilizing the LLM as the feature encoder, we utilize the prompts from TAPE [9], which allows us to assess the impact of LLM-based features on the model's performance. For datasets, we choose the widely used ogbn-arixv dataset, which contains paper title and abstract text information. Each node is an arXiv paper and each directed edge indicates that one paper cites another one. The task is to predict the 40 subject areas of arXiv CS papers, which are manually determined (i.e., labeled) by the paper’s authors and arXiv moderators. We split the data into three disjoint domains based on the publication year of the papers, i.e. 1950-2016, 2016-2018, 2018-2020. The statistical details for each domain are shown in the table below.
> > > >
> > > > | Domains | #Nodes | #Edge | Homo | #Avg Degree |
> > > > |:--:|:--:|:--:|:--:|:--:|
> > > > | 1950-2016 | 69,499 | 237,163 | 0.6945 | 3.41 |
> > > > | 2016-2018 | 51,241 | 111,754 | 0.6886 | 2.18 |
> > > > | 2018-2020 | 48,603 | 60,403 | 0.7092 | 1.24 |
> > > >
> > > > We explored two approaches to enhance the original node attributes:
> > > > 1. LLM enhanced text with word2vec embedding, which combines the title, abstract, and LLM-generated predictions and explanations into a single input. This composite text is then fed into word2vec. Then, the node features are obtained by averaging the embeddings of its combined input. We refer to it as arxiv-LLM-w2v.
> > > > 2. LLM enhanced text with BERT embedding, which combines the title, abstract, and LLM-generated predictions and explanations into a single input. This composite text is then fed into a pretrained DeBERTa. Then, the node features are obtained by sentence embedding. Note that, we did not finetune the DeBERTa like TAPE [9] paper, since we focus on unsupervised graph domain adaptation. We refer to it as arxiv-LLM-bert.
> > > >
> > > > First, we use MMD to characterize the degree of feature shift among these three datasets, i.e., ogbn-arxiv, arxiv-LLM-w2v and arxiv-LLM-bert. We consider 3 tasks, and the results are as follows:
> > > >
> > > > | source | target | ogbn-arxiv | arxiv-LLM-w2v | arxiv-LLM-bert |
> > > > |:--:|:--:|:--:|:--:|:--:|
> > > > | 1950-2016 | 2016-2018 | 0.0405 | 0.0400 $\downarrow$ | 0.0427 $\uparrow$ |
> > > > | 1950-2016 | 2018-2020 | 0.0528 | 0.0535 $\uparrow$ | 0.0796 $\uparrow$ |
> > > > | 2016-2018 | 2018-2020 | 0.0148 | 0.0138 $\downarrow$ | 0.0149 $\uparrow$ |
> > > >
> > > > As shown in the table, when word2vec is used to encode the LLM-enhanced text, the feature shift is reduced compared to the original node features. However, when BERT is used for encoding, the feature shift increases. This indicates that the choice of text encoding method significantly influences the degree of feature shift.

---

> > > > > ### Author Response · Authors · 2024-08-24
> > > > > **Response of Q2 (Part 2)**
> > > > >
> > > > > Next, we choose 5 recent graph domain adaptation models to assess the impact of LLM-based features on the model's performance. Each experiment is repeated 3 times, and we report the average Micro-F1 score with standard deviation as follows:
> > > > >
> > > > >
> > > > >
> > > > > | source | target | UDAGCN | AdaGCN | KBL | GRADE | A2GNN |
> > > > > |:--:|:--:|:--:|:--:|:--:|:--:|:--:|
> > > > > | | | |ogbn-arxiv | | | |
> > > > > | 1950-2016 | 2016-2018 | 52.42$\pm$0.52 | 61.78$\pm$0.18 | 50.88$\pm$0.24 |61.85$\pm$0.17 |60.18$\pm$0.54 |
> > > > > | 1950-2016 | 2018-2020 | 48.41$\pm$0.64 | 56.74$\pm$0.34 | 47.53$\pm$1.15 |57.19$\pm$0.26 |58.89$\pm$0.28|
> > > > > | 2016-2018 | 2018-2020 | 54.84$\pm$0.22 | 62.05$\pm$0.04 | 52.99$\pm$0.05 |61.42$\pm$0.14 |59.45$\pm$0.28|
> > > > > | | | |arxiv-LLM-w2v | | | |
> > > > > | source | target | UDAGCN | AdaGCN | KBL | GRADE | A2GNN |
> > > > > | 1950-2016 | 2016-2018 | 43.55$\pm$0.90 | 66.23$\pm$0.14 | 64.90$\pm$0.12 |65.41$\pm$0.31 |63.77$\pm$0.36 |
> > > > > | 1950-2016 | 2018-2020 | 39.56$\pm$0.54 | 62.04$\pm$0.29 | 60.67$\pm$0.29 |62.04$\pm$0.29 |65.35$\pm$0.12|
> > > > > | 2016-2018 | 2018-2020 | 47.20$\pm$0.99 | 68.42$\pm$0.04 | 66.77$\pm$0.13 |66.84$\pm$0.16 |65.83$\pm$0.40|
> > > > > | | | |arxiv-LLM-bert | | | |
> > > > > | source | target | UDAGCN | AdaGCN | KBL | GRADE | A2GNN |
> > > > > | 1950-2016 | 2016-2018 | 34.81$\pm$0.29 | 35.66$\pm$1.20 | 34.64$\pm$0.85 |35.04$\pm$0.86 |39.14$\pm$2.00 |
> > > > > | 1950-2016 | 2018-2020 | 30.18$\pm$0.25 | 31.69$\pm$0.42 | 30.26$\pm$1.09 |31.09$\pm$0.12 |35.40$\pm$1.55|
> > > > > | 2016-2018 | 2018-2020 | 39.20$\pm$2.53 | 39.93$\pm$1.90 | 37.27$\pm$2.06 |34.14$\pm$1.54 |43.62$\pm$1.87|
> > > > >
> > > > >
> > > > >
> > > > > As illustrated in the table, the performance of most baselines shows significant improvement with the arxiv-LLM-w2v dataset, whereas performance notably declines with the arxiv-LLM-bert dataset compared to the original ogbn-arxiv dataset. These results align with the MMD scores presented in the previous table, which indicate that arxiv-LLM-bert exhibits a larger distribution shift compared to the other datasets. This emphasizes the necessity of selecting an appropriate text encoder when utilizing LLM-enhanced text for graph domain adaptation. An effective choice of text encoder can greatly impact the performance and mitigate the distribution shifts in text-attributed graph domain adaptation tasks.

---

> > > > > > ### Author Response · Authors · 2024-08-24
> > > > > > **Response of Q4**
> > > > > >
> > > > > > **Q4.** *I think the transportation and financial ones are good examples. For social network, I dis-agree with this one as different domains have different label sets in most cases.*
> > > > > >
> > > > > > **A4.**  Thank you for your feedback on the examples for Q4. We agree that the previous social network examples may not have been entirely accurate. A more fitting application could be adapting influencer detection models from one social network (e.g., Instagram) to another (e.g., Facebook), where the goal is to categorize influencers into different tiers.
> > > > > > When dealing with different label spaces, open-set domain adaptation presents a viable solution. While this approach has been explored in the computer vision domain [7, 8], it remains underexplored in the graph domain. Investigating open-set UGDA could broaden the scope of domain adaptation, addressing a wider array of challenges. By doing so, we can develop algorithms that are not only more robust but also flexible enough to navigate the complexities and nuances of diverse, dynamic real-world scenarios. We will include these discussions in the final version as our future direction.
> > > > > >
> > > > > >
> > > > > > ---
> > > > > >
> > > > > > *Reference*
> > > > > >
> > > > > > [7] Balanced Open Set Domain Adaptation via Centroid Alignment
> > > > > >
> > > > > > [8] Subsidiary Prototype Alignment for Universal Domain Adaptation
> > > > > >
> > > > > > [9] Harnessing Explanations: LLM-to-LM Interpreter for Enhanced Text-Attributed Graph Representation Learning

---

> > > > > > > ### Comment · Reviewer_TozF · 2024-08-24
> > > > > > >
> > > > > > > Thanks for your response. I really appreciate the LLM experiments. I would like to see a full version of the experiment in the revision. According to the initial experimental results, I would like to raise my score.

---

> > > > > > > > ### Author Response · Authors · 2024-08-25
> > > > > > > > **Thanks for your reply!**
> > > > > > > >
> > > > > > > > Thanks for raising the score. We will include the full version of the experiment in the final version.

---

> ### Author Rebuttal · Authors · 2024-08-17
>
> **Q5.** *The paper does a good job of categorizing the basic UGDA scenario. It could be better to see a more comprehensive discussion on other related UGDA scenarios[3], multi-domain UGDA[5], and source-free UGDA[2,4].*
>
> **A5.** We are grateful for your suggestions regarding multi-domain and source-free UGDA.
> * Muti-domain UGDA extends the concept of domain adaptation to situations where there are multiple source domains and a single target domain.  This approach aims to learn a model that can generalize well across multiple source domains, and then adapt it to perform well on the target domain. Compared to standard UGDA, multi-domain UGDA can enhance generalization by leveraging the diversity of multiple source domains. However, it may require more complex models and additional computational resources.
> * Source-free UGDA advances domain adaptation by tackling the challenge of adapting models without access to labeled data from the source domains. This setting is more challenging as it involves learning to transfer knowledge without explicit supervision. Source-free UGDA methods often employ techniques such as self-training or consistency regularization to adapt the model to the target domain. Compared to UGDA, source-free UGDA may be more sensitive to domain shift and require careful selection of adaptation techniques.
>
> In summary, multi-domain UGDA and source-free UGDA are advanced techniques that aim to improve the model's robustness and generalization capabilities in complex and challenging scenarios. While these approaches offer promising solutions to the limitations of traditional single-domain UGDA, they also introduce new challenges and considerations that must be carefully addressed in practical applications. We will include a detailed discussion of multi-domain UGDA and source-free UGDA in our revised version.
>
> ---
>
> *Reference*
>
> [1] Demystifying Structural Disparity in Graph Neural Networks: Can One Size Fit All?
>
> [2] Collaborate to Adapt: Source-Free Graph Domain Adaptation via Bi-directional Adaptation
>
> [3] Graph Domain Adaptation: Challenges, Progress and Prospects
>
> [4] Source Free Graph Unsupervised Domain Adaptation
>
> [5] MSDS: A Novel Framework for Multi-Source Data Selection Based Cross-Network Node Classification
>
> [6] Pairwise Alignment Improves Graph Domain Adaptation

---

### Official Review · Reviewer_iZaq · 2024-07-20
**Revisiting, Benchmarking and Understanding Unsupervised Graph Domain Adaptation**

**Rating:** 6
**Confidence:** 3
**Correctness:** Yes
**Clarity:** Yes

**Review:**

Pros:
1. The paper introduces GDABench, which provides a standardized and comprehensive benchmarking framework for UGDA. 2. The use of five diverse datasets, covering various types of distribution shifts (feature shift, structure shift, and label shift), ensures that the benchmark is robust and applicable to a wide range of real-world scenarios.

Cons:
1. What are the computational requirements for running the experiments included in the benchmark, and how accessible are these requirements for researchers with limited resources? 2. How does the PyGDA library handle potential discrepancies in data preprocessing and experimental setups to ensure fair comparisons across different UGDA models? Some recent related works are missing[1-2]. 3. 4. Is the PyGDA library compatible with other popular graph analysis and machine learning tools, and are there plans to improve integration with these tools in the future? [1] GALA: Graph Diffusion-based Alignment with Jigsaw for Source-free Domain Adaptation [2] SPA: A Graph Spectral Alignment Perspective for Domain Adaptation

**Strengths:**

See Pros above.

**Additional Feedback:**

NA

**Documentation:**

NA

**Limitations:**

Yes

**Opportunities For Improvement:**

NA

**Relation To Prior Work:**

Yes

**Summary And Contributions:**

The paper presents GDABench, which includes 16 state-of-the-art UGDA algorithms tested across 5 diverse datasets and 74 adaptation tasks. The paper also introduces PyGDA, an easy-to-use library for training and evaluating UGDA methods, which promotes reproducibility and standardized comparisons in this research area.

---

> ### Author Rebuttal · Authors · 2024-08-17
>
> Dear Reviewer iZaq,
>
> Thank you for your thoughtful review and valuable feedback. We truly appreciate your positive evaluation of our paper. We address your concerns as follows.
>
> ---
> **Q1.** *What are the computational requirements for running the experiments included in the benchmark, and how accessible are these requirements for researchers with limited resources?*
>
> **A1.** Thanks for pointing this out. In Figure 6 of the Appendix, we've provided detailed information on the running time and memory consumption of the baseline algorithms.
> * For small scale dataset Airport (nodes below 5k), memory consumption ranges from 450 MB to 900 MB and running time within 20s.
> * For middle scale dataset ArnetMiner (nodes below 10k), memory consumption ranges from 2 GB to 30 GB and running time within 15min.
> * For large scale dataset MAG (nodes below 10w), memory consumption ranges from 2 GB to 80 GB and running time within 40min.
>
> It’s notable that several algorithms, such as A2GNN and CWGCN, achieve relatively good performance with less complexity. Most experiments can be conducted on a single NVIDIA Tesla A100 GPU with 32GB of memory, which is widely accessible for researchers, including those with limited resources.
>
> ---
> **Q2.** *How does the PyGDA library handle potential discrepancies in data preprocessing and experimental setups to ensure fair comparisons across different UGDA models?*
>
> **A2.** The PyGDA library has been designed with a universal approach to data processing and evaluation. To ensure consistency and fairness in comparisons across different UGDA methods, we have implemented unified APIs for commonly utilized modules. This modular structure enables researchers to align their preprocessing steps and experimental setups, facilitating fair comparisons even when using different UGDA models.
>
> ---
> **Q3.** *Some recent related works are missing.*
>
> **A3.** We appreciate your feedback regarding the inclusion of recent related works. We note that [1] focuses on source-free domain adaptation, which is beyond the scope of our current study, while [2] is tailored for IID image data and addresses the domain adaptation (DA) problem in terms of graph technique. Inspired by this two related works, we consider a broader discussion on DA problem and other related UGDA scenarios like multi-domain UGDA and source-free UGDA.
> * UDA vs UGDA: Unsupervised domain adaptation (UDA) involves the transfer of knowledge from a labelled source domain to an unlabelled target domain. A commonly used approach in domain adaptation is to minimize the domain discrepancy and learn domain-invariant representations, where great success has been achieved in computer vision and natural language processing communities. However, these methods assume that their inputs are independently and identically distributed data, thus they are not appropriate for tasks involving non-IID data, such as node classification in graph-structured data.
> * UGDA vs muti-domain UGDA: Muti-domain UGDA extends the concept of domain adaptation to situations where there are multiple source domains and a single target domain.  This approach aims to learn a model that can generalize well across multiple source domains, and then adapt it to perform well on the target domain. Compared to standard UGDA, multi-domain UGDA can enhance generalization by leveraging the diversity of multiple source domains. However, it may require more complex models and additional computational resources.
> * UGDA vs source-free UGDA: Source-free UGDA advances domain adaptation by tackling the challenge of adapting models without access to labeled data from the source domains. This setting is more challenging as it involves learning to transfer knowledge without explicit supervision. Source-free UGDA methods often employ techniques such as self-training or consistency regularization to adapt the model to the target domain. Compared to UGDA, source-free UGDA may be more sensitive to domain shift and require careful selection of adaptation techniques.
>
> ---
> **Q4.** *Is the PyGDA library compatible with other popular graph analysis and machine learning tools, and are there plans to improve integration with these tools in the future?*
>
> **A4.** The PyGDA library is designed to be compatible with most popular graph analysis and machine learning tools. We are actively working on improving its integration with frameworks such as DGL in future updates. We believe these improvements will significantly boost accessibility and usability for the research community.
>
> ---
>
> If our response have addressed your concerns, we kindly ask that you consider to increase one point for us!

---

> ### Author Response · Authors · 2024-08-24
>
> Dear Reviewer iZaq,
>
> We would like to express our sincere gratitude for the time and effort you have put into reviewing our paper and providing us with valuable feedback. We have tried our best to address your concerns.
>
> We kindly request that you take a few moments to review our rebuttal and let us know if there are any further concerns that we can address. We would appreciate your prompt response.
>
> Thank you once again for your feedback.

---

> > ### Author Response · Authors · 2024-08-28
> > **A gentle reminder for discussion**
> >
> > Dear Reviewer iZaq,
> >
> > As there are only three days remaining for rebuttal, we kindly request that you take a few moments to review our rebuttal and let us know if there are any further concerns that we can address. We would appreciate your prompt response.
> >
> > Thank you once again for your feedback.

---

> ### Author Response · Authors · 2024-08-31
> **A gentle reminder for discussion**
>
> Dear Reviewer iZaq,
>
> We sincerely appreciate the time and effort you’ve dedicated to reviewing our paper. As the discussion will end within **one day**, we kindly request that you take a few moments to review our rebuttal and let us know if there are any further concerns that we can address. We would appreciate your prompt response.

---

### Official Review · Reviewer_vbxq · 2024-08-01

**Rating:** 5
**Confidence:** 3
**Clarity:** Yes, the paper is well written.

**Review:**

Quality: The paper demonstrates a benchmark research with a comprehensive evaluation of UGDA methods. The introduction of PyGDA shows a commitment to the reproducible science in the field of graph domain adaptation.

Clarity: The paper appears to be well-structured and clearly presents its objectives, methodology, and findings.

Originality: The work presents a novel contribution to the field by providing a standardized benchmark and a practical tool  for UGDA research. This approach to systematizing the evaluation of UGDA methods is  valuable.

Pros:
1. Introduces a comprehensive benchmark for evaluating UGDA methods.
2. Provides a practical tool for implementing and evaluating UGDA methods.
4. Highlights the importance of addressing graph structural changes in domain adaptation.
5. Promotes reproducibility and standardization in UGDA research.

Cons:
1. Limited to node-level shifts, excluding graph-level shifts from the evaluation.
2. Does not fully explore all possible combinations of domain shift methods.
3. Lacks discussion on how findings might generalize to more complex or real-world scenarios.
4. Insufficient investigation of the impact of hyperparameters and model configurations on UGDA method performance.

**Strengths:**

- Comprehensive Benchmarking: GDABench provides a standardized platform for evaluating Understanding Unsupervised Graph Domain Adaptation methods, covering a wide range of datasets and tasks.

-  Analysis: The paper offers a thorough examination of various UGDA methods, highlighting the strengths and limitations of different approaches in handling distribution shifts.

- Practical Tool: The PyGDA library is a practical tool that simplifies the implementation and evaluation of UGDA methods, enhancing reproducibility and facilitating fair comparisons.

**Additional Feedback:**

N/A.

**Correctness:**

- The paper lacks an investigation of how different hyperparameters and model configurations affect the performance of UGDA methods.

**Documentation:**

Yes.

**Ethics:**

N/A.

**Limitations:**

Yes.

**Opportunities For Improvement:**

1. Limited Scope of Benchmark: The authors only consider node-level shifts, omitting graph-level shifts, which results in an incomplete evaluation.

2. Domain Shift Exploration: While the authors provide different domain shift methods for various datasets, the potential combinations of these methods could produce more domain shift scenarios that remain unexplored.

3. Limited Dataset Range: Although the benchmark includes a diverse set of datasets, the paper does not discuss how these findings might generalize to more complex or real-world scenarios that may involve more diverse types of graph data.

4. Insufficient Exploration of Hyperparameter Impact: The paper lacks an in-depth investigation of how different hyperparameters and model configurations affect the performance of UGDA methods, which could have provided deeper insights into model behavior.

**Relation To Prior Work:**

Yes.

**Summary And Contributions:**

This paper introduces GDABench, a comprehensive benchmark for evaluating the performance of Unsupervised Graph Domain Adaptation (UGDA) methods. The authors assess 16 UGDA algorithms on 5  datasets. The study reveals the performance among current UGDA models across different datasets and adaptation scenarios, highlighting the importance of addressing graph structural changes. Additionally, the paper introduces PyGDA, a library for training and evaluating UGDA methods, promoting reproducibility and standardization in the field.

---

> ### Author Rebuttal · Authors · 2024-08-17
>
> Dear Reviewer vbxq,
>
> We greatly appreciate the effort you have put into reviewing our manuscript. We address the reviewer’s concerns as follows:
>
> ---
>
> **Q1.** *Limited Scope of Benchmark: The authors only consider node-level shifts, omitting graph-level shifts, which results in an incomplete evaluation.*
>
> **A1.** Thanks for pointing this out. The majority of existing research in this area is specifically designed for node-level tasks and some methods cannot be adapted to graph-level tasks. For the remaining methods, we have added a pooling layer to evaluate their capabilities in graph-level domain adaptation. We employ three TUdatasets: Proteins, Mutagenicity, and Frankenstein, partitioning each dataset into 2 equally sized disjoint groups based on density shifts.
>
> |              | #Nodes | #Edges | #Feat | #Class | Num of graphs |
> | ------------ | ------ | ------ | ----- | ------ | ------------- |
> | Proteins     | ~39.06 | ~72.82 | 4     | 2      | 1113          |
> | Mutagenicity | ~30.32 | ~30.77 | 14    | 2      | 4337          |
> | Frankenstein | ~16.90 | ~17.88 | 780   | 2      | 4337          |
>
> The results are detailed in Tables 1 and 2 of the attached PDF.
> Among the methods, GRADE and A2GNN are domain adaptive message passing methods and the others are DA incorporated node embedding methods. Key observations are as follows:
> * **DA incorporated node embedding methods shows task-inconsisteny across node and graph-level tasks.** For example, DANE performs averagely in node-level tasks, but its performance improves significantly in graph-level tasks. This disparity highlights a challenge in predicting the performance of unsupervised graph domain adaptation (UGDA) models in real-world applications. The inconsistency suggests that models optimized for node-level tasks may not generalize well to graph-level tasks and vice versa. Consequently, this variability complicates the task of assessing how well these models will perform when deployed in diverse and complex real-world scenarios where both node-level and graph-level information may be critical.
> * **Domain adaptive message passing methods demonstrate superior and consistency performance across a wide range of datasets and tasks.**  As shown in Table 1&2 of attached pdf and Table 2&3 of manuscript, methods designed based on the inherent properties of GNN achieves the top-three best performance in 8 tasks out of 12 node-level tasks and top-two best performance in 5 tasks out of 6 graph-level tasks. This observation verified our findings that establishing domain adaptation principles by leveraging inherent properties of GNN can result in an effective and efficient approach to addressing the challenges of domain variability in graph datasets.
>
> To summarize, our observations underscore the importance of leveraging the intrinsic properties of GNNs to devise effective domain adaptation strategies, which not only enhances performance but also ensures consistency in real-world applications. Implementation details and parameters will be updated in the revised version.
>
> ---
> **Q2.** *Domain Shift Exploration: While the authors provide different domain shift methods for various datasets, the potential combinations of these methods could produce more domain shift scenarios that remain unexplored.*
>
> **A2.** We appreciate your point regarding the exploration of domain shift combinations. The 74 tasks compiled by the five carefully selected datasets can cover all combinations.
> * **Feature shift determined**: Tasks ES-PT, PT-ES, EN-DE, ED-EN, FR-ES, FR-PT, ES-FR, PT-FR, RU-ES, ES-RU, RU-PT, PT-RU, RU-FR and FR-RU in Twitch. Tasks JP-US, US-JP, JP-CN and CN-JP in MAG.
>
> * **Sturcture shift determined**: Tasks E-B, B-E, U-B, B-U, U-E and E-U in Airport. Tasks GP-DE and US-DE in MAG. Tasks FR-DE, RU-EN, RU-DE, ES-DE, EN-RU, DE-RU, DE-ES and DE-PT in Twitch.
>
> * **Lable shift determined**: Task FR-DE in MAG.
>
> * **Determined by both feature and structure shift**: Tasks D-A, D-C, A-D and C-D in ArmetMiner. Tasks FR-EN, EN-FR, PT-EN, EN-PT, DE-FR, FR-DE, PT-DE and EN-ES in Twitch. Tasks JP-FR, RU-PT, RU-CN and DE-JP in MAG.
>
> * **Determined by both feature and label shift**: Tasks EN-US, US-EN in MAG.
>
> * **Determined by both structure and label shift**: Tasks DE-US, FR-US, US-FR, FR-RU in MAG.
>
> * **All shifts effects**: Tasks B1-B2 and B2-B1 in Blog. Tasks A-C and C-A in ArnetMiner. Tasks DE-FR, CN-FR, JP-RU, RU-FR, CN-RU, FR-JP, RU-DE, CN-DE, RU-US, DE-CN, FR-CN, DE-RU and US-RU in MAG.

---

> ### Author Rebuttal · Authors · 2024-08-17
>
> **Q3.** *Limited Dataset Range: Although the benchmark includes a diverse set of datasets, the paper does not discuss how these findings might generalize to more complex or real-world scenarios that may involve more diverse types of graph data.*
>
> **A3.** Our benchmark includes a range of datasets with varying characteristics to capture different aspects of graph domain adaptation. This diversity aims to provide a broad perspective on the applicability of our methods. In real-world scenarios, applying graph adaptation methods effectively involves several key considerations: Firstly, it is imperative to develop tailored strategies specifically designed to address the structural shifts observed in graphs. For example, if a graph is dynamic and changing overtime, it is crucial to accord greater attention to its evolving structure. Secondly, recognizing the importance of the aggregation scope and aggregation architecture in GNNs' transferability within unsupervised graph domain adaptation (UGDA) are crucial. In real-world graphs, noise is inevitable, hence, strategically selecting effective neighbors not only improve performance but also avoid noise. Thirdly, by leveraging the properties of GNNs that make them inherently adaptable to changes in graph structure and data distribution, we can develop simple yet highly effective models.
>
> ---
>
> **Q4.** *Insufficient Exploration of Hyperparameter Impact: The paper lacks an in-depth investigation of how different hyperparameters and model configurations affect the performance of UGDA methods, which could have provided deeper insights into model behavior.*
>
> **A4.** Thank you for highlighting the importance of hyperparameters in UGDA performance. We investigate how various hyperparameters in common modules influence the performance of different UGDA methods on ArnetMiner dataset (task D->A). We focus on two key aspects: the number of GNN layers and the representation dimensions. Results are shown in Fig.1 in the attached pdf.
>
> * Impact of representation dimensions: All models demonstrate some degree of sensitivity to the node representation dimensions, highlighting the importance of selecting an appropriate dimensionality for optimal performance. Some models (like UDAGCN, StruRW, and A2GNN) may have a relatively narrow optimal dimension range, others (like CWGCN, JHGDA and SepcReg) continue to benefit from increased dimensionality. Overall, the optimal dimensions are highly likely to be within [64, 128, 256].
>
> * Impact of GNN layers: Most models show less sensitivity to the number of layers. While some may show steeper decrease with increasing layers  (like UDAGCN, StruRW, and A2GNN) for issues like overfitting or vanishing/exploding gradients. In most cases, less than 3 layer will be better.
>
> Overall, our analysis reveals that both the number of GNN layers and the representation dimensions play important roles in determining the performance of UGDA methods. The optimal choice of these hyperparameters depends on the specific dataset and adaptation task, highlighting the need for careful hyperparameter tuning and model selection when applying UGDA methods in practice. We will add the exploration of hyperparameter impact on other datasets in the revised version.
>
> ---

---

> ### Author Response · Authors · 2024-08-24
>
> Dear Reviewer vbxq,
>
> We would like to express our sincere gratitude for the time and effort you have put into reviewing our paper and providing us with valuable feedback. We have tried our best to address your concerns.
>
> We kindly request that you take a few moments to review our rebuttal and let us know if there are any further concerns that we can address. We would appreciate your prompt response.
>
> Thank you once again for your feedback.

---

> ### Author Response · Authors · 2024-08-28
> **A gentle reminder for discussion**
>
> Dear Reviewer vbxq,
>
> As there are only three days remaining for rebuttal, we kindly request that you take a few moments to review our rebuttal and let us know if there are any further concerns that we can address. We would appreciate your prompt response.
>
> Thank you once again for your feedback.

---

> ### Author Response · Authors · 2024-08-31
> **A gentle reminder for discussion**
>
> Dear Reviewer vbxq,
>
> We sincerely appreciate the time and effort you’ve dedicated to reviewing our paper. As the discussion will end within **one day**, we kindly request that you take a few moments to review our rebuttal and let us know if there are any further concerns that we can address. We would appreciate your prompt response.

---

### Author Response · Authors · 2024-08-24
**Global Response**

We sincerely appreciate the insightful and constructive feedback from all the reviewers.

We have addressed all the questions of reviewers in each individual response. According to the suggestions of reviewers, we also conduct additional experiments on graph level domain adaptation and LLM embedding for text-attributed graphs. We would like to highlight the following points:
- We have updated our PyGDA library to support graph level domain adaptation task. To perform a graph-level domain adaptation task, only one parameter is added to the model by setting `mode='graph'`.
- As suggested by Reviewer vbxq, we have conducted additional experiments on graph level domain adaptation. The scripts for reproducing our graph level domain adaptation task are available at https://github.com/pygda-team/pygda/tree/main/benchmark/graph.
- As suggested by Reviewer TozF, we have investigated whether the distribution gap narrows after utilizing the LLM as the feature encoder on the ogbn-arixv datasets. We explored two approaches to enhance the original node attributes. The results emphasize the necessity of selecting an appropriate text encoder when utilizing LLM-enhanced text for graph domain adaptation.
- The scripts for reproducing our LLM embedding for text attributed graph domain adaptation are available at https://github.com/pygda-team/pygda/tree/main/benchmark/llm.

We hope the updated results and our responses have addressed the questions that reviewers have. Please let us know if there are any other questions or suggestions.

Thanks!

---

### Decision · Program_Chairs · 2024-09-26

**Decision:**

Accept (Poster)

**Comment:**

This paper introduces GDABench, a comprehensive benchmark for evaluating the performance of Unsupervised Graph Domain Adaptation (UGDA) methods. After repeated communication between the author and the reviewers, the relevant doubts and uncertainties were resolved. Ultimately, based on various factors, including innovation and rigor, I have determined that this paper can be accepted.